



# Using NDII patterns to constrain semi-distributed rainfall-runoff models in tropical nested catchments

Nutchanart Sriwongsitanon[1,2], Wasana Jandang[1,2], Thienchart Suwawong[1,2] and Hubert H.G. Savenije[3]

[1] Department of Water Resources Engineering, Faculty of Engineering, Kasetsart University, Bangkok, Thailand
[2] Remote Sensing Research Centre for Water Resources Management (SENSWAT), Faculty of Engineering, Kasetsart University, Bangkok, Thailand
[3] Delft University of Technology, Stevinweg 1, 2600 GA Delft, The Netherlands

*Correspondence to*: Nutchanart Sriwongsitanon (fengnns@ku.ac.th)

**Abstract.** A parsimonious semi-distributed rainfall-runoff model has been developed for flow prediction. In distribution,
attention is paid to both timing of runoff and heterogeneity of moisture storage capacities within sub-catchments. This model
is based on the lumped FLEXL model structure, which has proven its value in a wide range of catchments. To test the value
of distribution, the gauged Upper Ping catchment in Thailand has been divided into 32 sub-catchments, which can be
grouped into 5 gauged sub-catchments where internal performance is evaluated. To test the effect of timing, firstly excess
rainfall was calculated for each sub-catchment, using the model structure of FLEXL. The excess rainfall was then routed to
its outlet using the lag time from storm to peak flow (*TlagF*) and the lag time of recharge from the root zone to the
groundwater (*TlagS*), as a function of catchment size. Subsequently, the Muskingum equation was used to route sub-
catchment runoff to the downstream sub-catchment, with the delay time parameter of the Muskingum equation being a
function of channel length. Other model parameters of this semi-distributed FLEX-SD model were kept the same as in the
calibrated FLEXL model of the entire Upper Ping basin, controlled by station P.1 located at the centre of Chiang Mai
Province. The outcome of FLEX-SD was compared to: 1) observations at the internal stations; 2) the calibrated FLEXL
model; and 3) the semi-distributed URBS model - another established semi-distributed rainfall-runoff model. FLEX-SD
showed better or similar performance both during calibration and especially in validation. Subsequently, we tried to
distribute the moisture storage capacity by constraining FLEX-SD on patterns of the NDII (normalized difference infrared
index). The readily available NDII appears to be a good proxy for moisture stress in the root zone during dry periods. The
maximum moisture holding capacity in the root zone is assumed to be a function of the maximum seasonal range of NDII
values, and the annual average NDII values to construct 2 alternative models: FLEX-SD-NDII$_{Max-Min}$ and FLEX-SD-NDII$_{Avg}$,
respectively. The additional constraint on the moisture holding capacity by the NDII improved both model performance and
the realism of the distribution. Distribution of Sumax using annual average NDII values was found to be well correlated with
the percentage of evergreen forest in 31 sub-catchments. Spatial average NDII values were proved to be highly corresponded
with the root zone soil moisture of the river basin, not only in the dry season but also in the water limited ecosystem. To
check how well the model represents root zone soil moisture, the performance of the FLEX-SD-NDII model was compared
to time series of the soil wetness index (SWI). The correlation between the root zone storage and the daily SWI appeared to

be very good, even better than the correlation with the NDII, because NDII does not provide good estimates during wet periods. The SWI, which is partly model-based, was not used for calibration, but appeared to be an appropriate index for validation.

# 1 Introduction

Runoff is one of the most important components of the hydrological cycle and can be monitored by the installation of a gauging station. Unfortunately, there are only a limited number of high quality gauging stations available due to topographic, financial and human resources limitations. A wide variety of rainfall-runoff models have been developed in gauged and ungauged catchments in different parts of the globe. Most rainfall-runoff models are categorised as lumped models, which can provide runoff estimates only at the site of calibration. These models include FLEXL, FLEX-Topo (Euser et al., 2015;
Gao et al., 2014), NAM (Bao et al., 2011; Tingsanchali and Gautam, 2000; Vaitiekuniene, 2005; Yew Gan et al., 1997), SCS (Hawkins, 1990; Lewis et al., 2000; Mishra et al., 2005; Suresh Babu and Mishra, 2011; Yahya et al., 2010), and many others.

To alleviate the limitation of lumped-rainfall-runoff models, URBS was developed as a semi-distributed nonlinear rainfall
runoff routing model, which can account for the spatial and temporal variation in rainfall by separating a catchment into a series of sub-catchments (Mapiam and Sriwongsitanon, 2009). Therefore, URBS claims to provide runoff estimates not only at a gauging station but also at any required upstream location (Carroll, 2004; Malone, 1999). URBS has been applied successfully for real time flood forecasting in a range of catchments from small to very large basins in Australia and in many countries worldwide (Malone, 2006; Malone et al., 2003; Mapiam and Sriwongsitanon, 2009; Mapiam et al., 2014;
Rodriguez et al., 2005; Sriwongsitanon, 2010). However, this model only addresses the distribution of travel times and does not address the effect of distributed storage capacities that affect the partitioning of moisture and hence the water balance.

Sriwongsitanon et al. (2016) proposed to use the NDII as a proxy for root zone soil moisture and showed its effectiveness in 8 sub-catchments of the Upper Ping river basins in Thailand. This is in agreement with the study carried out by Castelli et al.
(2019) who found reasonable correlations between Landsat 7 NDII values and measured root-zone soil moisture contents of rainfed olive trees growing in the arid regions of south eastern Tunisia, supporting the use of NDII as a proxy for soil water content in arid regions.

Mao and Liu (2019) developed the Water And Ecosystem Simulator (WAYS) which is a distributed model based on FLEXL
to simulate discharge as well as root zone water storage (RZWS) on a global scale across 10 major basins comprising Congo, Nile, Niger, Yangtze, Ganges, Parana, Amazon, Mississippi, Murray-Darling, and Mekong. The model showed a good performance in simulating evaporation and discharge. It also could simulate RZWS in most of the regions through

comparison with NDII (correlation (r) ranging from 0.951 and 0.713 with an average of 0.883). This is with the exception of some basins such as the Amazon, Murray-Darling and Mississippi (r ranging from 0.552 and 0.677) which have high percentage of forest areas, trees intercepting deep groundwater (e.g. Eucalyptus) or plenty of precipitation with low moisture stress where NDII may not correctly reflect RZWS dynamics.

Regarding the use of the Soil Water Index (SWI), which is partly model-based, as a proxy for root zone soil moisture, Paulik et al. (2014) found reasonable correlations between in-situ soil moisture data from 664 stations - available through the international Soil Moisture Network (ISMN) - and the SWI produced from ASCAT SSM estimates. The average of Pearson correlation coefficients was shown to be 0.54, with 64.4% of all time series greater than 0.5. SWI may be used as another

index for the soil moisture state of a basin with or without moisture stress.

Among the wide range of existing lumped-rainfall-runoff models, FLEXL has proven to be an adequate model for runoff estimation in a wide range of catchments (Fenicia et al., 2011; Fenicia et al., 2008; Gao et al., 2014; Kavetski and Fenicia, 2011; Tekleab et al., 2015). This model was further developed by Gharari et al. (2011) and Gao et al. (2016) to account for

the spatial variability of landscape characteristics (FLEX-TOPO), useful for prediction in ungauged basins (Savenije, 2010). Moreover, Sriwongsitanon et al. (2016) demonstrated that catchment-scale soil moisture content in the rootzone of vegetation computed from FLEXL is correlated with the remotely sensed Normalized Difference Infrared Index (NDII), as a proxy for the equivalent water thickness (EWT) in the root zone, especially during periods of moisture stress.

This study aims to utilize the fundamental model structure of FLEXL, include distributed time lags and channel routing as used in URBS, and include distributed root zone soil moisture capacity per sub-catchment so as to create a new parsimonious semi-distributed FLEX model for flood and flow monitoring within the (ungauged) sub-catchments of the gauged Upper Ping River Basin. Distribution of time lags is expected to improve hydrograph shape, particularly the timing and shape of the peaks, which would improve best-fit parameters, but it does not affect the partitioning of the hydrological fluxes or the water

balance. Since the root zone storage is the main control on flux partitioning, the distribution of the root zone moisture storage capacity would potentially have a larger impact on model performance. Therefore, the spatial variation of the NDII, as an indicator of root zone moisture stress, has been used to distribute moisture storage capacities among sub-catchments, while the model-based SWI, as an estimator of moisture storage, was used for validation.

The main steps undertaken in the following sections are the following:

1. To introduce the effect of runoff timing in a catchment with multiple sub-catchments, the travel times to the outfall of each individual sub-catchment are computed on the basis of topographical indicators and the routing of the discharge from the sub-catchment outfall to stations further downstream are computed using the Muskingum method. These time lags are then applied both in the FLEX-SD model system and in the well-established URBS model, for the purpose of comparison.





These two semi-distributed models only account for timing, but not for the distribution of the moisture storage capacity, a crucial parameter in runoff generation.

2. Subsequently, the effect of distribution of the root zone moisture storage is studied in the FLEX-SD model, making use of the spatial distribution pattern of the maximum and minimum range of NDII values, and the annual average NDII values to

construct 2 alternative models: FLEX-SD-NDII$_{Max-Min}$ and FLEX-SD-NDII$_{Avg}$, respectively.

3. Finally, as a validation of the model and to check if the models are capable of representing the internal moisture states, the simulated root zone moisture storage is compared to the independent data set of the Soil Wetness Index (SWI).

## 2 Study area and datasets

### 2.1 Study area

The Upper Ping River Basin (UPRB) is situated between latitude 17∘14′30′′ to 19∘47′52′′N and longitude 98∘ 4′30′′ to 99∘22′30′′E in the provinces of Chiang Mai and Lam Phun. The catchment area of the basin is approximately 25,370 km$^2$. The basin is dominated by well-forested, steep mountains in a generally north-south alignment (Sriwongsitanon and Taesombat, 2011). The areal average annual rainfall and runoff of the basin from 2001-2016 are 1,224 mm/yr and 235

mm/yr, respectively. The land use for the UPRB in 2013 can be classified into 6 main classes comprising forest, irrigated agriculture, rainfed agriculture, bare land, water body, and others, which cover approximately 77.40%, 3.11%, 12.54%, 1.99%, 1.23%, and 3.73% of the catchment area, respectively (Land Development Department, LDD). The landform of the UPRB varies from an undulating to a rolling terrain with steep hills at elevations of 1500– 2000 m, and valleys of 330–500m (Mapiam and Sriwongsitanon, 2009; Sriwongsitanon, 2010). Chiang Dao district, north of Chiang Mai is the origin of the

Ping River, which flows downstream to the south to become the inflow of the Bhumibol Dam – a large dam with an active storage capacity of about 9.7 billion m$^3$ (Sriwongsitanon, 2010). The climate of the basin is dominated by tropical monsoons. The southwest monsoon causes a rainy season between May and October and the northeast monsoon brings dry weather and low temperatures between November and April.   Only 6,142 km$^2$ of the total area controlled by the runoff station P.1 (situated at the centre of Chiang Mai) is selected for this study (Fig. 1). The catchment area of the station P.1 is divided into

32 sub-catchments (Fig. 1) where the semi-distributed rainfall-runoff models are tested.

### 2.2 Rainfall data

Daily rainfall data from 48 non-automatic rain-gauge stations located within the UPRB and its surroundings from 2001-2016 were used in this study. These data are owned and operated by the Thai Meteorological Department and the Royal Irrigation Department. These data have been validated for their accuracy on monthly basis using double mass curve and some

inaccurate data were removed from the time series before spatially averaging using an inverse distance square (IDS) to be applied as the forcing data of URBS, FLEXL, and FLEX-SD. Mean areal rainfall depth for each of 32 sub-catchments varies





between 1,100 (S17) and 1,402 (S11) mm/yr as shown in Figure 1 (b) while the average rainfall depth of P.1 is approximately 1,224 mm/yr.

## 2.3 Runoff data

The Royal Irrigation Department (RID) operates 7 daily runoff stations in the study area between 2001 and 2016 as shown in

Fig. 1. Catchment P.56A was rejected from the study because it is located upstream of Mae Ngat reservoir. Outflow data from the reservoir were used as input data in model calibration. Runoff data at the remaining 6 stations were used for the study since they are not affected by large reservoirs. The data have been checked for their accuracy by comparing them with average rainfall data covering their catchment areas at the same periods. Table 1 presents the catchment characteristics and hydrological data for these 6 gauging stations in the UPRB. In this study, the catchments of these 6 stations were divided

into 32 sub-catchments (see Fig. 1) with areas ranging from 57 to 230 km². High variation of catchment size is due to the proximity between the locations of these runoff stations and the outlets of the tributaries. Runoff data have been checked for their accuracy by comparing the annual runoff coefficient between all stations. The comparison revealed that the runoff coefficients at P.20 in 2006 and 2011 are overestimated, while the runoff coefficient at P.21 in 2004 is underestimated and in 2007 and 2009 are overestimated due to incorrect rating curves (see Fig. 2). These inaccurate data would affect the results of

model calibration.

## 2.4 NDII Data

The Normalized Difference Infrared Index (NDII) is a ratio of the near-infrared (NIR) and shortwave infrared (SWIR) bands, centred at 859 and 1,640 nm, respectively, as shown in Eq. (1). In this study, the NDII was calculated using the MODIS level 3 surface reflectance product (MOD09A1), which is available at 500 m resolution in an 8-day composite of the gridded level

2 surface reflectance products. Atmospheric correction has been carried out to improve the accuracy and can be downloaded from ftp://e4ftl01.cr. usgs.gov/MOLT (Vermote et al., 2011). The 8 day NDII values between 2002-2016 were averaged over each of 31 sub-catchments of the UPRB to be used for estimating model parameter within sub-catchment and to be compared to the 8 day average $S_u$ (root zone storage) values extracted from the model results at each station.

$$NDII = \frac{(NIR - SWIR)}{(NIR + SWIR)} \tag{1}$$

## 2.5 SWI Data

The near real-time Soil Water Index (SWI) is derived from the reprocessed Surface Soil Moisture (SSM) data derived from the ASCAT sensor (Brocca et al., 2011; Paulik et al., 2014), which is a C-Band Scatterometer measuring at a frequency of 5.255 GHz in VV-polarisation (Paulik et al., 2014). The product makes use of a two-layer water balance model to describe the time series relationship between surface and profile soil moisture. This dataset of moisture conditions is available on a

daily basis for eight characteristic time windows 1, 5, 10, 15, 20, 40, 60 and 100 days. The global scale SWI dataset is

available at 0.1 degree, which is about 10 km resolution, within 3 days after observation and can be downloaded from the Copernicus Global Land Service website. The dataset is available from January 2007 onwards. Since the SWI dataset is not complete in 2007, only the data between 2008 and 2016 were used in this study.

## 3 Theoretical background

### 3.1 FLEXL model

FLEXL is a lumped hydrological model comprising five reservoirs: a snow reservoir ($Sw$), an interception reservoir ($Si$), an unsaturated soil reservoir ($Su$), a fast-response reservoir ($Sf$), and a slow-response reservoir ($Ss$) (Gao et al., 2014). Excess rainfall from a snow reservoir, an interception reservoir, and an unsaturated soil reservoir is divided and routed into a fast-response reservoir and a slow-response reservoir using two lag functions. It includes the lag time from storm to peak flow ($TlagF$) and the lag time of recharge from the root zone to the groundwater ($TlagS$). Each reservoir has process equations that connect the fluxes entering or leaving the storage compartment to the storage in the reservoirs (so-called constitutive functions) (Sriwongsitanon et al., 2016). The water balance equations and constitutive equations for each conceptual reservoir are summarised in Fig. 3 and Table 2. The total number of model parameters is 11. Forcing data include daily average rainfall and potential evaporation derived by the Penman-Monteith equation.

### 3.1.1 Snow reservoir

The snow routine, not very relevant in Thailand, can play an important role in areas with snow. When there is snow cover and the temperature ($T_i$) is above $Tt$, the effective precipitation is equal to the sum of rainfall ($P_i$) and snowmelt ($M_i$). The snowmelt ($M_i$) is calculated by the melted water per day per degree Celsius above $Tt$ ($F_{DD}$) (Eq. 2). The snow reservoir uses the water balance equation, Eq. (3), where $Sw_i$ (mm) is the storage of the snow reservoir.

### 3.1.2 Interception reservoir

Interception is more important in summer and autumn. The interception evaporation $Ei_i$ was calculated by potential evaporation ($Ep_i$) and the storage in the interception reservoir ($Si_i$), with a daily maximum storage capacity ($Imax$) (Eqs. 4, 5). The interception reservoir uses the water balance equation, Eq. (6), presented in Table 2.

### 3.1.3 Root zone reservoir

The root zone routine, which is the core of the hydrological models, determines the amount of runoff generation. In this study, we applied the widely used beta function of the Xinanjiang model (Ren-Jun, 1992) to compute the runoff coefficient for each time step as a function of the relative soil moisture. In Eq. (7), $Cr_i$ indicates the runoff coefficient, $Su_i$ is the storage in the root zone reservoir, $Sumax$ is the maximum moisture holding capacity in the root zone and $\beta$ is the parameter



describing the spatial process heterogeneity of the runoff threshold in the catchment. In Eq. (8), $Pe_i$ indicates the effective rainfall and snowmelt into the root zone routine; $Ru_i$ represents the generated flow during rainfall events. In Eq. (9) $Su_i$, $Sumax$ and potential evaporation ($Ep_i$) were used to determine actual evaporation from the root zone $Ea_i$; $Ce$ indicates the fraction of $Sumax$ above which the actual evaporation is equal to potential evaporation, here set to 0.5 as previously

suggested by Savenije (1997) otherwise $Ea_i$ is constrained by the water available in $Su_i$. The unsaturated soil reservoir uses the water balance equation, Eq. (10), presented in Table 2.

### 3.1.4 Fast response reservoir

In Eq. (11), $Rf_i$ indicates the flow into the fast-response routine; $D$ is a splitter to separate recharge from preferential flow. Equations (12) and (13) were used to describe the lag time between storm and peak flow. $Rf_{t-i+1}$ is the generated fast runoff in

the unsaturated zone at time $t - i + 1$, $TlagF$ is a parameter which represents the time lag between storm and fast runoff generation, $c_{lagF}(i)$ is the weight of the flow in $i - 1$ days before and $Rfl_i$ is the discharge into the fast-response reservoir after convolution.

A linear-response reservoir, representing a linear relationship between storage and release, was applied to conceptualize the discharge from the surface runoff reservoir, fast response reservoirs and slow-response reservoirs. In Eq. (14), $Qff_i$ is the

surface runoff, with timescale $Kff$, active when the storage of the fast-response reservoir exceeds the threshold $Sfmax$. In Eq. (15), $Qf_i$ represents the fast runoff; $Sf_i$ represents the storage state of the fast response reservoirs; $Kf$ is the timescales of the fast runoff. The fast response reservoir uses the water balance equation, Eq. (16), presented in Table 2.

### 3.1.5 Slow response reservoir

In Eq. (17), $Rs_i$ indicates the recharge of the groundwater reservoir. Equations (18) and (19) were used to describe the lag

time of recharge from the root zone to the groundwater. $Rs_{t-i+1}$ is the generated slow runoff in the groundwater zone at time $t - i + 1$, $TlagS$ is a parameter which represents the lag time of recharge from the root zone to the groundwater, $c_{lagS}(i)$ is the weight of the flow in $i - 1$ days before and $Rsl_i$ is the discharge into the slow-response reservoir after convolution. In Eq. (20) , $Qs_i$ represents the slow runoff; $Ss_i$ represents the storage state of the groundwater reservoir; $Ks$ is the timescales of the slow runoff. The slow response reservoir uses the water balance equation, Eq. (21), presented in Table 2.

### 3.2 URBS model

URBS was developed by Queensland Department of Natural Resources and Mines in 1990 based on the structures of RORB (Laurenson and Mein, 1990) and WBNM (Boyd et al., 1987). URBS is a semi-distributed rainfall-runoff model that can provide runoff estimates not only at the calibrated station but also at the outlet of every sub-catchment at any required location upstream. The calibrated catchment area needs to be divided into sub-catchments to obtain different areal rainfall

and different catchment and channel travelling time.

Table 3 presents 5 main processes used in URBS comprising the calculation of the initial loss, proportional loss, excess rainfall, catchment routing and channel routing. Excess rainfall is calculated separately between pervious and impervious areas. For the pervious area, URBS assumes that there is the maximum initial loss rate ($IL_{max}$) to be reached before any rainfall becoming the effective rainfall ($R_i^{eff}$). The initial loss ($IL_i$) can be recovered when the rainfall rate ($R_i$) is less than

the recovering loss rate ($rlr$) per time interval ($\delta t$) (see Eq. (22)).

Excess rainfall for each time step is calculated using Eq. (23) by weighting the excess rainfall between pervious and impervious area using a ratio of the cumulative infiltration ($F_i$) and the maximum infiltration capacity ($F_{max}$). The recovering rate is included by simply reducing the amount infiltrated after every time step using the reduction coefficient ($k_{\delta t}$) as shown

in Eq. (24), and the pervious excess rainfall ($R_i^{per}$) is calculated using the Eq. (25), where $pr$ is the proportional runoff coefficient. The remaining water ($1-pr)R_i^{eff}$ will infiltrate to the root zone storage ($dF_i$) (see Eq. (26)). Excess rainfall is then routed to the centroid of any sub-catchment using a nonlinear reservoir relationship ($S_i = KQ_i^m$). The parameter $m$ is the catchment non-linearity and $K$ is the catchment travel time, which can be calculated for different sub-catchment using the multiplication between the catchment lag time coefficient ($\beta$) and square root of each sub-catchment area ($A$) (see Eq. (27)).

Thereafter, the outflow at the centroid of each sub-catchment is routed along a reach downstream of each sub-catchment using the Muskingum equation ($S_i^{ch} = K_{ch}(XI_i+(1-X)Q_i)$). The parameter $X$ is the Muskingum coefficient and $K_{ch}$ is the channel travel time, which can be calculated for different sub-catchment using the multiplication between the channel lag coefficient ($\alpha$) and the reach length (L) between the closest location in the channel to the centroid and the outlet of each sub-

catchment (see Eq. (28)).

## 4 Methodology

### 4.1 Development of the semi-distributed FLEX model

The first step in distribution is to account for the timing of floods and the rooting of flood waves as a function of topographical factors. The resulting semi-distributed FLEX-SD model therefore is expected to better represent the shape of

hydrographs, although it would not affect the partitioning of fluxes or the water balance. The root zone storage capacity is a strong control on partitioning, affecting both runoff generation and evaporation. Therefore, distribution of this parameter would potentially affect overall model performance more strongly than merely the timing of the peaks. Therefore, in a second step, the NDII, as a proxy for moisture storage, is used to assess the distribution of moisture storage among sub-catchments.





### 4.1.1 Accounting for distributed timing and channel-routing

FLEX-SD is set-up by applying lumped models for each sub-catchment, adding up to a semi-distributed model for a downstream calibration site. Therefore, the catchment area of any gauging station needs to be divided into sub-catchments. Runoff estimates at each sub-catchment can be simulated using the structure of the original FLEXL by calculating different

excess rainfall for each sub-catchment. The excess rainfall of each sub-catchment is routed to its outlet using the lag time from

rainfall to surface runoff ($TlagF$) and the lag time of recharge from the root zone to the groundwater ($TlagS$). In this study, $TlagF$ and $TlagS$ are calculated in hours instead of days to increase model performance. The lag time is distributed among sub-catchments using the following equations.

$$TlagF_{sub} = TlagF\sqrt{A_{sub}/A} \qquad (29)$$
$$TlagS_{sub} = TlagS\sqrt{A_{sub}/A} \qquad (30)$$

where, $Tlag$ is a lag time parameter for the entire catchment of a calibrated gauging station. The lag time of each sub-catchment ($Tlag_{sub}$) is scaled by the square root of each sub-catchment area divided by the overall catchment area ($A$).

Runoff estimates from an upstream sub-catchment is later routed from its outlet to the outlet of a downstream sub-catchment

using the Muskingum method (Eq. (31)) before adding to the runoff estimates of the downstream sub-catchment.

$$S_{chnl-sub} = K_{sub}\left(XQ_{up} + (1-X)Q_{down}\right) \qquad (31)$$
$$K_{sub} = \alpha L_{sub} \qquad (32)$$

where, $\alpha$ and $X$ are the delay time parameter and the channel routing parameter for the entire catchment, respectively. The delay time parameter of each sub-catchment ($K_{sub}$) can be calculated by the multiplication between $\alpha$ and the main channel

length of each sub-catchment as shown in Equation (32).

### 4.1.2 Accounting for distributed root zone storage at sub-catchment scale using the maximum and minimum values of NDII (FLEX-SD-NDIIMaxMin model)

The Normalized Difference Infrared Index (NDII) was used to estimate root zone storage capacity for each sub-catchment. The NDII values, which are available at 8 day intervals, were found to correlate well with the 8-day average root zone

moisture content ($Su$) simulated by FLEXL during the dry period in eight sub-catchments in the UPRB (Sriwongsitanon et al., 2016). The relation between NDII and $Su$ can be described by an exponential function of the type: $ae^{b(NDII)}+c$, with $c$ close to zero. The maximum value that $Su$ can achieve is $Sumax$, the storage capacity of the root zone. The hypothesis is that the ecosystem creates sufficient storage to overcome a critical period of drought (Gao et al., 2014; Savenije and Hrachowitz, 2017). Every year has a maximum range of storage variation. If a sufficiently long NDII record is available, then the

maximum of the annual ranges of the NDII should provide an estimate of the root zone storage capacity $Sumax$. By calibrating the hydrological FLEX model to discharge observations at the gauging stations, for each gauged catchment a





*Sumax* value can be calibrated. This is a representative *Sumax* value for a particular gauging station, consisting of n sub-areas, indicated by $Sumax_n$.

$$Sumax_n = \frac{\sum_{i=1}^{n}(A_i Sumax_i)}{\sum_{i=1}^{n} A_i} \tag{33}$$

By using the NDII as proxy for root zone storage, we have developed the following equation for the proxy root zone storage capacity $Sumax'_i$ for a sub-area within a river basin consisting of 31 sub-catchments:

$$Sumax'_i = \frac{\left[e^{b \times NDII_{i,max}} - e^{b \times NDII_{i,min}}\right]_{max}}{\left[e^{b \times NDII_{n,max}} - e^{b \times NDII_{n,min}}\right]_{max}} \tag{34}$$

Where $Sumax'_i$ is a scaled proxy for the root zone storage capacity of each sub-catchment, and *b* is the remaining calibration parameter, because the constant *c* and the factor *a* of the exponential function drop out. The $NDII_{i,max}$ and $NDII_{i,min}$ represent the maximum and minimum values of NDII for each year of each sub-catchment, while the $NDII_{n,max}$ and $NDII_{n,min}$ indicate the maximum and minimum values of NDII for each year in the reference basin, in this case, the entire Upper Ping basin controlled by station P.1. The unscaled root zone storage capacity per sub-catchment then becomes:

$$Sumax_i = Sumax_n \frac{Sumax'_i}{Sumax'_n} \tag{35}$$

Where $Sumax_n$ is the calibrated value of the root zone storage capacity of the gauged catchment, and $Sumax'_n$ is the area weighted proxy for the root zone storage capacity.

$$Sumax'_n = \frac{\sum_{i=1}^{n}(A_i Sumax'_i)}{\sum_{i=1}^{n} A_i} \tag{36}$$

### 4.1.3 Accounting for distributed root zone storage at sub-catchment scale using average value of NDII (FLEX-SD-NDII$_{Avg}$)

Instead of applying the maximum and minimum of the annual ranges of the NDII to distribute root zone storage at a sub-catchment scale, we tested the annual average NDII value of each sub-area to calculate $Sumax'_i$ as presented in the following equation.

30 $$Sumax'_i = \left(0.5 - \frac{R}{2}\right) + R\left(\frac{(e^{b \times NDII_i}) - (e^{b \times NDII_{i \rightarrow n}})_{min}}{(e^{b \times NDII_{i \rightarrow n}})_{max} - (e^{b \times NDII_{i \rightarrow n}})_{min}}\right) \tag{37}$$





Where $NDII_i$ represents the annual average NDII value of each sub-catchment, while $(e^{b \times NDII_{i \to n}})_{max}$ and $(e^{b \times NDII_{i \to n}})_{min}$ indicate the maximum and minimum values of exponential function produced by the annual average NDII value within 32 sub-catchments. The parameters $b$ and $R$ can be determined by model calibration. The parameter $R$ is suggested to vary

between 0.2 and 0.8 to force a scaled factor $Sumax'_i$ to be more than 0 and less than 1. The average NDII value is supposed to reflect the maximum moisture storage capacity as well, since a high maximum value also leads to a higher average, but is much easier to calculate. However, this method requires the introduction of the additional calibration parameter $R$.

## 4.2 Applications of URBS, FLEXL, FLEX-SD, FLEX-SD-NDII<sub>Max-Min</sub> and FLEX-SD-NDII<sub>Avg</sub>

URBS, FLEX-SD, FLEX-SD-NDII<sub>Max-Min</sub> and FLEX-SD-NDII<sub>Avg</sub> were calibrated (2001-2011) and validated (2012-2016) at P.1 station located in the city of Chiang Mai. Since these models are semi-distributed rainfall-runoff models, they can provide runoff estimates in any required locations upstream of P.1 station, resulting in runoff estimates for P.4A, P.20, P.21, P.75 and P.67. As benchmarks for analysis, the calibrated FLEXL model was also used to estimate runoff at these 5 stations. In addition, all semi-distributed models were calibrated and validated at these 5 stations for a fair comparison with the results

of the locally calibrated FLEXL model at each internal station (presented in Annex A). The model parameters of the calibrated models were determined using the MOSCEM-UA (Multi-Objective Shuffled Complex Evolution Metropolis-University of Arizona) algorithm(Vrugt et al., 2003) by finding the Pareto-optimal solutions defined by three objective functions of the Kling-Gupta Efficiencies for high flows, low flows, and the flow duration (KGE<sub>E</sub>, KGE<sub>L</sub> and KGE<sub>E</sub>), respectively. KGE<sub>E</sub> is analysed using the following equations, where $\bar{X}$ is the average observed discharge, $\bar{Y}$ is the average

simulated discharge, $S_X$ is the standard deviation of observed discharge, $S_Y$ is the standard deviation of simulated discharge, and r is the linear correlation between observations and simulations. KGE<sub>L</sub> can be calculated using the logarithm of flows to emphasize low flows. The Nash-Sutcliffe Efficiency (NSE) is an independent statistical indicator, which is not utilised in the objective function but merely used to summarised model performance. The model calculates at daily time steps, but this is disaggregated to hourly to take into account the time lags. The output is again aggregated to daily time steps.

$$KGE = 1 - ED \tag{38}$$

$$ED = \sqrt{(r-1)^2 + (\alpha - 1)^2 + (\beta - 1)^2} \tag{39}$$

$\quad \alpha = S_Y / S_X \tag{40}$

$$\beta = \bar{Y} / \bar{X} \tag{41}$$



# 5 Results

## 5.1 Accuracy of runoff estimates simulated by URBS, FLEXL, FLEX-SD, FLEX-SD-NDII$_{Max-Min}$ and FLEX-SD-NDII$_{Avg}$

The performance criteria of the runoff estimates simulated by URBS, FLEXL, FLEX-SD, FLEX-SD-NDII$_{Max-Min}$ and FLEX-
SD-NDII$_{Avg}$ calibrated on P.1 runoff data for the period (2001-2011), and for the validation period (2012-2016), are presented in Table 4 and Table 5, respectively. Model parameters of these models are presented in Table A1. Figures 4, 5 and 6 present the output of the five models for all stations compared to observations, as accumulated flows, hydrographs on logarithmic scale, and duration curves, respectively. For comparison, the results of the calibrated and validated models at each of the 6 individual stations are presented in Figures A1, A2 and A3. Of course, these calibrated models close the water
balance better, but this may be due to over-fitting. Table 4 shows that FLEXL, FLEX-SD, FLEX-SD-NDII$_{MaxMin}$ and FLEX-SD-NDII$_{Avg}$ calibrated at P.1 produce similar overall accuracy with an average NSE of 0.73, 0.73, 0.72 and 0.75, respectively during the calibration period, while URBS obtained a lower NSE of 0.68. However, FLEXL acquired higher KGE values compared to other models. Table 5 surprisingly shows that FLEXL attains the lowest NSE value of 0.53 during validation, compared to NSE values of 0.70, 0.68, 0.67 and 0.65 produced by FLEX-SD-NDII$_{Avg}$, FLEX-SD, FLEX-SD-
NDII$_{Max-Min}$, and URBS, respectively. It should be realized that FLEXL was calibrated individually for each sub-catchment, while the other models were used in predictive mode. The fact that in the validation mode all semi-distributed models obtain more accurate results than the lumped and calibrated FLEXL model indicates a higher predictive capacity of the semi-distributed models.

Figure 4 clearly shows that the distributed models are not capable of closing the water balance in four stations except at P.1 and P.67 located in the main Ping. While FLEXL, which is calibrated at each individual station can mimic the pattern, this may be due to over-fitting. In P.75, the models over-estimate the observed flow. This is due to flow regulation and water withdrawals in the managed parts of the sub-catchments (there is the large Mae Ngad dam upstream of P.75). The duration curves in Figure 6 confirm this and also show that the observed lowest flow is below the modelled flow, almost throughout.
This is likely due to water abstractions for urban and agricultural water supply. In contrast, the models underestimate the flows in P.21 and P.20, which are intensively used catchments with rating problems. On the other hand, the flow is over-estimated at P.4A, which drains a mountainous catchment with evergreen forest. The lumped models are apparently not yet capable to distinguish well between these different landscapes. A landscape-based model as suggested by Gharari et al. (2011) and Savenije (2010) could be the next step for improvement.

The model parameters used in FLEXL, FLEX-SD and FLEX-SD-NDII$_{Max-Min}$ and FLEX-SD-NDII$_{Avg}$ are summarized in Table A1. The SD model provides different values for *TlagF* (the time lag between storm and fast runoff generation), and *TlagS* (the lag time of recharge from the root one to the groundwater); the other parameters are kept the same as the

calibrated values for P.1. Since *TlagF* and *TlagS* were designed to be related to the catchment area, the parameter values for each station are more reasonable compared to the values given by FLEXL. It can be noted that the values of *TlagF* obtained by FLEX-SD-NDII$_{Avg}$ are much closer to the ones presented by FLEX-SD compared to the values obtained by FLEX-SD-NDII$_{Max-Min}$.

The *Sumax* values generated by FLEX-SD-NDII$_{Avg}$ and FLEX-SD-NDII$_{MaxMin}$ are quite different between the 31 sub-catchments. A large part of the landscape within the Upper Ping River Basin is covered by evergreen forest which may affect the soil moisture of each sub-catchment. The relationships between the percentage of evergreen forest and *Sumax* in 31 sub-catchments calibrated and validated by FLEX-SD-NDII$_{Avg}$ and FLEX-SD-NDII$_{MaxMin}$ are presents in Figure A4. The figure

displays quite high $R^2$ correlation of 0.69 introduced by FLEX-SD-NDII$_{Avg}$ compared to a small $R^2$ value of 0.01 exhibits by FLEX-SD-NDII$_{MaxMin}$. It seems that FLEX-SD-NDII$_{Avg}$ introduces more realistic *Sumax* values in forested landscapes.

In general, the FLEX-SD-NDII models provides lower *Sumax* estimates than the other models, constraining evaporation in the dry season (which provides more realistic recessions in Figure 5), but compensates for this reduction by a smaller $\beta$

value, so as to limit excessive flood generation. Since these parameters jointly control Eq. (7), they can compensate for each other, leading to equifinality. If one of the parameters is constrained by additional information, as is the case here using the NDII, then this is no longer possible. The performance with respect to best fit parameters may reduce in the process, but the model has gained realism and hence predictive power.

We see that the FLEXL-SD-NDII models show the highest realism (illustrated clearly in Figure 5 and Figure A2 in the appendix A) but not a very good performance in the sub-catchment P.20, although still better than the other SD models. P.20 remains a difficult sub-catchment to predict due to its flow regulation and water consumption. Also we see that adding constraints to model calibration does not always improve best-fit performance, as compared to free calibration, but that realism can be improved. To further test the realism of the models, in the following section the outputs of the models are

compared to observations of NDII and the global scale SWI dataset for verification.

## 5.2 The relationship between the average root zone soil moisture storage (*Su$_i$*) and the average NDII and SWI

Sriwongsitanon et al. (2016) suggested that NDII can be used as a proxy for soil moisture storage in hydrology. Therefore, the 8 day average NDII values were compared to the 8 day average root zone moisture storage (*Su$_i$*) as calculated by FLEXL, FLEX-SD, FLEX-SD-NDII$_{Max-Min}$ and FLEX-SD-NDII$_{Avg}$. Table 6 shows the coefficients of the exponential relationships

and the coefficients of determination ($R^2$) together with the NSE for the wet season, and the dry season for all six stations. The table shows that the time series of NDII values correlate well with *Su* values during the dry season by giving $R^2$ value (average of all sub-catchments) of 0.75, 0.76, 0.79, and 0.78 for the 4 models respectively. The NSE value given by these models are 0.50, 0.53, 0.57, and 0.58, respectively. During the wet season these correlations are much worse, resulting in





average $R^2$ value of 0.43, 0.41, 0.46, and 0.46 respectively, while the NSE value are 0.44, 0.49, 0.46, and 0.48, respectively. The same procedure was also carried out for all 31 sub-catchments and the results shown in Table 7 which indicates that during the dry season, the average $R^2$ value produced by FLEX-SD, FLEX-SD-NDII$_{Max-Min}$ and FLEX-SD-NDII$_{Avg}$ are 0.71, 0.74, and 0.74, respectively, and NSE are 0.41, 0.45, and 0.52, respectively. During the wet season, $R^2$ value are 0.36, 0.41,

and 0.41, respectively, and NSE are 0.38, 0.36, and 0.40, respectively. FLEX-SD-NDII$_{Avg}$ provides the highest $R^2$ and NSE values for all seasons. Detailed information for 31 sub-catchments presents in Table A2. The Table confirms that FLEX-SD-NDII$_{Avg}$ performs slightly better than FLEX-SD-NDII$_{MaxMin}$, but this is not surprising as it has one more calibration parameter ($R$), which provides an additional degree of freedom.

It is to be noted that some sub-catchments show much lower $R^2$ and NSE values compared to the rest, which may be the result of land use/land cover. The evergreen forest probably experiences less moisture stress compared to other land use/land cover, in which situation the NDII does not relate as well to root zone soil moisture. Therefore, Figure A5 displays the dry season relationships between percent of evergreen forest and NSE values from the relationships between the average scaling NDII values and simulated root zone moisture storage (Su) in 31 sub-basins calibrated and validated by FLEX-SD, FLEX-

SD-NDII$_{MaxMin}$, FLEX-SD-NDII$_{Avg}$. The figure obviously shows that $R^2$ provided by these models are quite high with the values of 0.85, 0.52, and 0.80, respectively. The results indicate that the relationship with the average root zone soil moisture storage is affected by the ecology of the river basin. It should be noted that the NSE values contributed by FLEX-SD-NDII$_{Avg}$ for 31 sub-catchments are generally higher than those of produced by FLEX-SD-NDII$_{MaxMin}$, and especially by FLEX-SD. The results confirm the power of NDII to capture the spatial variation of root zone soil moisture within the sub-

catchment scale. Figure A6 presents the corresponding scatter plots for six stations and it clearly shows that the correlation is much better in the dry season than in the wet season. This is not surprising, as it was argued by Sriwongsitanon et al. (2016) that the relation between NDII and root zone soil moisture can only be observed by this remote sensing product when the vegetation is experiencing moisture stress. Hence correlations between root zone soil moisture and NDII are poor during the wet season. Because the FLEX-SD-NDII was constrained by the spatial variability of NDII ranges, the good correlation

between $Su$ and NDII during the dry season may not be surprising. Therefore, an additional test was done, testing the modelled $Su$ values at daily time step with the daily SWI values, for all models.

Table 8 and Figure A7 shows that the time series of SWI40 correlates well with $Su$ values during the dry season by giving $R^2$ value of 0.86, 0.89, 0.87, and 0.88 simulated by FLEXL, FLEX-SD, FLEX-SD-NDII$_{MaxMin}$, FLEX-SD-NDII$_{Avg}$ respectively,

and NSE value of 0.76, 0.79, 0.81, and 0.81, respectively. During the wet season these correlations are in the same order of magnitude as in the dry season with the average $R^2$ value of 0.87, 0.88, 0.89, and 0.88, respectively, with NSE value of 0.80, 0.84, 0.83, and 0.83, respectively. The results reveal that seasons and model types do not influence the Su-SWI relationship. All FLEX models, essentially using the same runoff generation procedure, have shown their ability to simulate Su in correspondence with SWI.



Detailed information for 31 sub-catchments is displayed in Table 9 and Table A3. The correlation does not significantly deviate among different models for all seasons. We also show the time series plots of the average NDII (Scaling), average SWI (Scaling) and the average root zone moisture storage (Su) calculated by all models for six runoff stations in the wet and

the dry seasons separately in Figures A8 and A9, respectively. One should realise, however, that the SWI is partly model based and that this may affect the good correspondence during the wet season. It can therefore be concluded that the NDII is a suitable parameter to constrain hydrological models during moisture recession, but that it works less well under wet conditions. The SWI, being partly model-based, is less attractive as a model constraint, but does not suffer from a drawback during wet conditions, and hence serves well as an assessment criterion, particularly during wet conditions. As a result, the

NDII appears to be useful to constrain hydrological models during dry conditions and both SWI and NDII appear to be useful to test model performance and to assess moisture states of river basins.

## 6 Conclusion

Most lumped rainfall-runoff models are controlled by a gauging station at the outfall on which it is calibrated. Runoff estimation at any location upstream requires indirect approaches such as model parameter transfer from gauged stations to

ungauged locations, or applying relationships between model parameters and catchment characteristics to the ungauged locations. By using any of these approaches, uncertainty in runoff estimation for ungauged catchments is unavoidable. A semi-distributed hydrological model could offer a better alternative. Besides taking into account lag times and flood routing (as in FLEX-SD), it has been shown that it is required to account for the spatial variation of the moisture holding capacity of the root zone. Therefore, the model was constrained by using NDII patterns as a proxy for the spatial variation of root zone

moisture leading to distributed $Sumax$-values among sub-catchments. We concluded that the maximum of a series of annual ranges ($NDII_{MaxMin}$) and annual average ($NDII_{Avg}$) of NDII values offers an effective proxy for estimating the appropriate $Sumax$ values in the different sub-catchments. It was shown that the two FLEX-SD-NDII models significantly improved the relationship between NDII and the modelled root zone moisture storage ($Su_i$) of all 31 sub-catchments. Moreover, the time series of the SWI correlated very well with the modelled root zone moisture storage ($Su_i$) of all sub-basins controlled by

runoff stations.

The model parameters provided by the semi-distributed FLEX models are more realistic compared to the original FLEXL since they are distributed according to catchment characteristics comprising catchment area, reach length, and remote sensing indices (NDII and SWI). A next step in the analysis is to account for diversity in landscape composition and related

model structures among sub-catchments (Gao et al., 2016), which would allow for a distinction between the main rainfall-runoff mechanisms belonging to different landscape types. This study confirms the result of the earlier study by Sriwongsitanon el al. (2016) who concluded that NDII can be used as a proxy for catchment-scale root zone moisture deficit





when plants are exposed to water stress. However, during the wet season when soil moisture is replenished as a result of rainfall, NDII values are no longer well correlated with soil moisture. However, the – partly model-based – SWI proved to be a reliable index to estimate soil moisture both under water stressed and wet conditions.

## Appendix A

Table A1 to A3

Fig. A1 to Fig. A9

## Acknowledgements

The authors would like to express our sincere gratitude to Faculty of Engineering, Kasetsart University for financially supporting this research. We are also indebted to the Royal Irrigation Department and Thai Meteorology Department for

providing the hydrological data.

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





**Table 1: Catchment characteristics and hydrological data for 6 gauging stations in the study area**

| Runoff Station | P.20 | P.75 | P.4A | P.67 | P.21 | P.1 |
|---|---|---|---|---|---|---|
| Area (km$^2$) | 1,309 | 3,029 | 1,954 | 5,333 | 516 | 6,142 |
| Altitude range (m) | 993 | 1,035 | 686 | 1,058 | 581 | 1,067 |
| Length main channel (km) | 89 | 126 | 143 | 155 | 52 | 185 |
| Average channel slope | 0.006 | 0.005 | 0.004 | 0.004 | 0.01 | 0.004 |
| Average rainfall (mm/yr) | 1,227 | 1,250 | 1,176 | 1,221 | 1,220 | 1,224 |
| Rainfall Range (mm/yr) | 926 – 1,640 | 900 – 1,643 | 829 – 1,449 | 866 – 1,570 | 728 – 1,606 | 847 – 1,565 |
| Average runoff (mm/yr) | 324.8 | 233.6 | 186.6 | 229.2 | 261.8 | 235.2 |
| Runoff Range (mm/yr) | 94.2 – 672.4 | 67.0 – 480.1 | 37.3 – 455.2 | 34.0 – 495.5 | 80.2 - 522.4 | 54.2 – 494.1 |
| Irrigated Area (%) | 15.7 | 18.1 | 9.4 | 15.1 | 17.4 | 15 |
| Evergreen Forest (%) | 10.2 | 9.6 | 39.7 | 20.0 | 22.1 | 19.8 |
| Forest Area (%) | 76.0 | 74.0 | 82.1 | 76.1 | 67.8 | 73.9 |
| % Runoff Average | 25.9 | 18.2 | 15.1 | 18 | 20.7 | 18.5 |
| % Runoff Range | 10.2 - 51.7 | 7.4 - 34.2 | 4.5 - 31.4 | 3.9 - 34.6 | 11.0 - 32.5 | 6.4 - 33.9 |





**Table 2: Constitutive and water balance equations used in FLEXL**

| No. | Reservoir | Constitutive equations | Equation | Water balance equations | Equation |
|---|---|---|---|---|---|
| 1 | Snow | $M_i = \begin{cases} F_{DD}(T_i\text{-}T_t) \ ; \ T_i > T_t \\ 0 \qquad\quad ; \ T_i \leq T_t \end{cases}$ | (2) | $\dfrac{dSw}{dt} = Ps_i\text{-}M_i$ | (3) |
| 2 | Interception | $Ei_i = \begin{cases} Ep_i \ ; \ Si_i > 0 \\ 0 \quad ; \ Si_i = 0 \end{cases}$ <br> $Ptf_i = \begin{cases} 0 \quad ; \ Si_i < Imax \\ Pr_i \ ; \ Si_i \geq Imax \end{cases}$ | (4) <br><br> (5) | $\dfrac{dSi}{dt} = Pr_i\text{-}Ei_i\text{-}Ptf_i$ | (6) |
| 3 | Unsaturated soil | $Cr_i = 1\text{-}\left(1\text{-}\dfrac{Su_{i\text{-}1}}{Sumax}\right)^{\beta}$ <br> $Ru_i = Pe_iCr_i$ <br> $Ea_i = (Ep_i\text{-}Ei_i)\min\left(\dfrac{Su_i}{Sumax\cdot Ce},1\right)$ | (7) <br><br> (8) <br><br> (9) | $\dfrac{dSu}{dt} = Pe_i(1\text{-}Cr_i)\text{-}Ea_i$ | (10) |
| 4 | Fast response | $Rf_i = Ru_iD$ <br> $c_{lagF}(j) = \dfrac{j}{\sum_{u=1}^{TlagF} u}$ <br> $Rfl_i = \sum_{j=1}^{TlagF} C_{lagF}(j) \cdot Rf_{i\text{-}j\text{-}1}$ <br> $Qff_i = \dfrac{max(0,Sf_i\text{-}Sfmax)}{Kff}$ <br> $Qf_i = \dfrac{Sf_i}{Kf}$ | (11) <br><br> (12) <br><br> (13) <br><br> (14) <br><br> (15) | $\dfrac{dSf}{dt} = Rfl_i\text{-}Qff_i\text{-}Qf_i$ | (16) |
| 5 | Slow response | $Rs_i = Ru_i(1\text{-}D)$ <br> $c_{lagS}(j) = \dfrac{j}{\sum_{u=1}^{TlagS} u}$ <br> $Rsl_i = \sum_{j=1}^{TlagS} C_{lagS}(j) \cdot Rs_{i\text{-}j\text{-}1}$ <br> $Qs_i = \dfrac{Ss_i}{Ks}$ | (17) <br><br> (18) <br><br> (19) <br><br> (20) | $\dfrac{dSs}{dt} = Rs_i\text{-}Qs_i$ | (21) |



**Table 3: Constitutive equations used in URBS**

| Processes | Constitutive Equations | Equation |
|---|---|---|
| Initial Loss | $IL_i = \begin{cases} IL_{i-1} & ; R_{i-1} > rlr.\delta t \\ IL_{i-1} + rlr.\delta t - R_{i-1} & ; R_{i-1} \leq rlr.\delta t \\ IL_{max} & ; IL_{i-1} > IL_{max} \end{cases}$ | (22) |
| Proportional Loss and Excess Rainfall | $R_i^E = \dfrac{F_i}{F_{max}} C_{imp} R_i + \left(1 - \dfrac{F_i}{F_{max}}\right) R_i^{per}$ | (23) |
| | $F_i = k_{\delta t} F_{i-1} + dF_i$ | (24) |
| | $R_i^{per} = pr(R_i^{eff})$ | (25) |
| | $dF_i = (1-pr)R_i^{eff}$ | (26) |
| Catchment Routing | $S_i = \beta\sqrt{A}Q_i^m$ | (27) |
| Channel Routing | $S_i^{ch} = \alpha L(XI_i + (1-X)Q_i)$ | (28) |





**Table 4: Statistical indicators at each station for calibration period provided by FLEXL and semi-distributed models. Best performance underlined.**

| Station | Model | Calibration period (2001 - 2011) | | | | | | | |
|---|---|---|---|---|---|---|---|---|---|
| | | Statistical indicators for calibrate at each station | | | | Statistical indicators for calibrate at station P.1 | | | |
| | | NSE | KGE$_E$ | KGE$_L$ | KGE$_F$ | NSE | KGE$_E$ | KGE$_L$ | KGE$_F$ |
| P.20 | (1) URBS | 0.59 | 0.79 | 0.30 | 0.91 | 0.58 | 0.63 | 0.49 | 0.70 |
| | (2) FLEXL | 0.66 | 0.82 | 0.50 | 0.96 | 0.66 | 0.82 | 0.50 | 0.96 |
| | (3) FLEX-SD | 0.66 | 0.83 | 0.51 | 0.96 | 0.62 | 0.59 | 0.38 | 0.63 |
| | (4) FLEX-SD-NDII$_{Max-Min}$ | 0.67 | 0.83 | 0.65 | 0.98 | 0.59 | 0.50 | 0.50 | 0.54 |
| | (5) FLEX-SD-NDII$_{Avg}$ | 0.67 | 0.82 | 0.73 | 0.96 | 0.64 | 0.67 | 0.40 | 0.72 |
| P.75 | (1) URBS | 0.76 | 0.87 | 0.81 | 0.96 | 0.68 | 0.81 | 0.78 | 0.87 |
| | (2) FLEXL | 0.73 | 0.86 | 0.65 | 0.97 | 0.73 | 0.86 | 0.65 | 0.97 |
| | (3) FLEX-SD | 0.79 | 0.89 | 0.82 | 0.97 | 0.77 | 0.87 | 0.83 | 0.93 |
| | (4) FLEX-SD-NDII$_{Max-Min}$ | 0.80 | 0.90 | 0.84 | 0.98 | 0.80 | 0.88 | 0.82 | 0.94 |
| | (5) FLEX-SD-NDII$_{Avg}$ | 0.79 | 0.90 | 0.83 | 0.98 | 0.75 | 0.82 | 0.84 | 0.86 |
| P.4A | (1) URBS | 0.64 | 0.82 | 0.63 | 0.98 | 0.64 | 0.79 | 0.57 | 0.89 |
| | (2) FLEXL | 0.71 | 0.84 | 0.71 | 0.93 | 0.71 | 0.84 | 0.71 | 0.93 |
| | (3) FLEX-SD | 0.71 | 0.85 | 0.58 | 0.95 | 0.68 | 0.75 | 0.65 | 0.79 |
| | (4) FLEX-SD-NDII$_{Max-Min}$ | 0.71 | 0.84 | 0.61 | 0.91 | 0.65 | 0.70 | 0.62 | 0.74 |
| | (5) FLEX-SD-NDII$_{Avg}$ | 0.70 | 0.84 | 0.67 | 0.93 | 0.74 | 0.83 | 0.68 | 0.90 |
| P.67 | (1) URBS | 0.72 | 0.86 | 0.73 | 0.97 | 0.77 | 0.84 | 0.70 | 0.90 |
| | (2) FLEXL | 0.76 | 0.87 | 0.75 | 0.95 | 0.76 | 0.87 | 0.75 | 0.95 |
| | (3) FLEX-SD | 0.80 | 0.90 | 0.72 | 0.96 | 0.82 | 0.87 | 0.70 | 0.91 |
| | (4) FLEX-SD-NDII$_{Max-Min}$ | 0.78 | 0.88 | 0.72 | 0.95 | 0.83 | 0.86 | 0.71 | 0.89 |
| | (5) FLEX-SD-NDII$_{Avg}$ | 0.79 | 0.89 | 0.72 | 0.96 | 0.83 | 0.87 | 0.71 | 0.90 |
| P.21 | (1) URBS | 0.64 | 0.82 | 0.48 | 0.95 | 0.60 | 0.77 | 0.53 | 0.84 |
| | (2) FLEXL | 0.70 | 0.85 | 0.88 | 0.98 | 0.70 | 0.85 | 0.88 | 0.98 |
| | (3) FLEX-SD | 0.74 | 0.86 | 0.82 | 0.93 | 0.61 | 0.78 | 0.37 | 0.86 |
| | (4) FLEX-SD-NDII$_{Max-Min}$ | 0.73 | 0.87 | 0.85 | 0.97 | 0.61 | 0.76 | 0.45 | 0.85 |
| | (5) FLEX-SD-NDII$_{Avg}$ | 0.72 | 0.86 | 0.73 | 0.95 | 0.66 | 0.74 | 0.38 | 0.80 |
| P.1 | (1) URBS | 0.80 | 0.90 | 0.76 | 0.97 | 0.80 | 0.90 | 0.76 | 0.97 |
| | (2) FLEXL | 0.82 | 0.90 | 0.76 | 0.98 | 0.82 | 0.90 | 0.76 | 0.98 |
| | (3) FLEX-SD | 0.86 | 0.93 | 0.75 | 0.97 | 0.86 | 0.93 | 0.75 | 0.97 |
| | (4) FLEX-SD-NDII$_{Max-Min}$ | 0.87 | 0.93 | 0.77 | 0.98 | 0.87 | 0.93 | 0.77 | 0.98 |
| | (5) FLEX-SD-NDII$_{Avg}$ | 0.87 | 0.93 | 0.77 | 0.99 | 0.87 | 0.93 | 0.77 | 0.99 |
| Average | (1) URBS | 0.69 | 0.84 | 0.62 | 0.96 | 0.68 | 0.79 | 0.64 | 0.86 |
| | (2) FLEXL | 0.73 | 0.86 | 0.71 | 0.96 | 0.73 | 0.86 | 0.71 | 0.96 |
| | (3) FLEX-SD | 0.76 | 0.87 | 0.70 | 0.96 | 0.73 | 0.80 | 0.61 | 0.85 |
| | (4) FLEX-SD-NDII$_{Max-Min}$ | 0.76 | 0.87 | 0.74 | 0.96 | 0.72 | 0.77 | 0.64 | 0.82 |
| | (5) FLEX-SD-NDII$_{Avg}$ | 0.76 | 0.87 | 0.74 | 0.96 | 0.75 | 0.81 | 0.63 | 0.86 |





**Table 5: Statistical indicators at each station for validation period provided by FLEXL and semi-distributed models. Best performance underlined.**

| Station | Model | Validation period (2012 - 2016) | | | | | | | |
|---|---|---|---|---|---|---|---|---|---|
| | | Statistical indicators for calibrate at each station | | | | Statistical indicators for calibrate at station P.1 | | | |
| | | NSE | KGE$_E$ | KGE$_L$ | KGE$_F$ | NSE | KGE$_E$ | KGE$_L$ | KGE$_F$ |
| P.20 | (1) URBS | 0.44 | 0.80 | 0.31 | 0.90 | 0.72 | 0.66 | 0.52 | 0.70 |
| | (2) FLEXL | 0.43 | 0.82 | 0.52 | 0.92 | 0.43 | 0.82 | 0.52 | 0.92 |
| | (3) FLEX-SD | 0.46 | 0.83 | 0.52 | 0.92 | 0.77 | 0.59 | 0.39 | 0.62 |
| | (4) FLEX-SD-NDII$_{Max-Min}$ | 0.50 | 0.83 | 0.64 | 0.94 | 0.76 | 0.47 | 0.52 | 0.53 |
| | (5) FLEX-SD-NDII$_{Avg}$ | 0.49 | 0.83 | 0.74 | 0.92 | 0.74 | 0.67 | 0.42 | 0.71 |
| P.75 | (1) URBS | 0.70 | 0.84 | 0.79 | 0.96 | 0.72 | 0.81 | 0.76 | 0.87 |
| | (2) FLEXL | 0.33 | 0.74 | 0.60 | 0.91 | 0.33 | 0.74 | 0.60 | 0.91 |
| | (3) FLEX-SD | 0.79 | 0.89 | 0.81 | 0.97 | 0.76 | 0.87 | 0.82 | 0.93 |
| | (4) FLEX-SD-NDII$_{Max-Min}$ | 0.78 | 0.88 | 0.82 | 0.97 | 0.78 | 0.87 | 0.81 | 0.94 |
| | (5) FLEX-SD-NDII$_{Avg}$ | 0.78 | 0.88 | 0.81 | 0.98 | 0.73 | 0.82 | 0.82 | 0.86 |
| P.4A | (1) URBS | 0.55 | 0.75 | 0.72 | 0.98 | 0.55 | 0.74 | 0.62 | 0.88 |
| | (2) FLEXL | 0.58 | 0.77 | 0.70 | 0.93 | 0.58 | 0.77 | 0.70 | 0.93 |
| | (3) FLEX-SD | 0.59 | 0.78 | 0.66 | 0.94 | 0.46 | 0.68 | 0.64 | 0.79 |
| | (4) FLEX-SD-NDII$_{Max-Min}$ | 0.59 | 0.78 | 0.68 | 0.91 | 0.41 | 0.63 | 0.62 | 0.73 |
| | (5) FLEX-SD-NDII$_{Avg}$ | 0.55 | 0.77 | 0.69 | 0.93 | 0.59 | 0.76 | 0.69 | 0.90 |
| P.67 | (1) URBS | 0.65 | 0.83 | 0.76 | 0.96 | 0.71 | 0.82 | 0.73 | 0.90 |
| | (2) FLEXL | 0.51 | 0.78 | 0.76 | 0.92 | 0.51 | 0.78 | 0.76 | 0.92 |
| | (3) FLEX-SD | 0.70 | 0.85 | 0.75 | 0.95 | 0.71 | 0.83 | 0.72 | 0.90 |
| | (4) FLEX-SD-NDII$_{Max-Min}$ | 0.69 | 0.84 | 0.76 | 0.93 | 0.69 | 0.81 | 0.73 | 0.88 |
| | (5) FLEX-SD-NDII$_{Avg}$ | 0.67 | 0.83 | 0.74 | 0.95 | 0.71 | 0.82 | 0.74 | 0.89 |
| P.21 | (1) URBS | 0.54 | 0.78 | 0.49 | 0.94 | 0.50 | 0.72 | 0.50 | 0.84 |
| | (2) FLEXL | 0.66 | 0.82 | 0.88 | 0.98 | 0.66 | 0.82 | 0.88 | 0.98 |
| | (3) FLEX-SD | 0.67 | 0.83 | 0.80 | 0.93 | 0.61 | 0.76 | 0.36 | 0.85 |
| | (4) FLEX-SD-NDII$_{Max-Min}$ | 0.65 | 0.83 | 0.84 | 0.97 | 0.60 | 0.74 | 0.45 | 0.85 |
| | (5) FLEX-SD-NDII$_{Avg}$ | 0.65 | 0.82 | 0.72 | 0.95 | 0.64 | 0.73 | 0.38 | 0.80 |
| P.1 | (1) URBS | 0.68 | 0.84 | 0.74 | 0.97 | 0.68 | 0.84 | 0.74 | 0.97 |
| | (2) FLEXL | 0.66 | 0.85 | 0.71 | 0.97 | 0.66 | 0.85 | 0.71 | 0.97 |
| | (3) FLEX-SD | 0.74 | 0.88 | 0.73 | 0.97 | 0.74 | 0.88 | 0.73 | 0.97 |
| | (4) FLEX-SD-NDII$_{Max-Min}$ | 0.75 | 0.88 | 0.74 | 0.98 | 0.75 | 0.88 | 0.74 | 0.98 |
| | (5) FLEX-SD-NDII$_{Avg}$ | 0.76 | 0.88 | 0.75 | 0.98 | 0.76 | 0.88 | 0.75 | 0.98 |
| Average | (1) URBS | 0.59 | 0.81 | 0.63 | 0.95 | 0.65 | 0.76 | 0.65 | 0.86 |
| | (2) FLEXL | 0.53 | 0.80 | 0.70 | 0.94 | 0.53 | 0.80 | 0.70 | 0.94 |
| | (3) FLEX-SD | 0.66 | 0.84 | 0.71 | 0.95 | 0.68 | 0.77 | 0.61 | 0.84 |
| | (4) FLEX-SD-NDII$_{Max-Min}$ | 0.66 | 0.84 | 0.75 | 0.95 | 0.67 | 0.73 | 0.64 | 0.82 |
| | (5) FLEX-SD-NDII$_{Avg}$ | 0.65 | 0.84 | 0.74 | 0.95 | 0.70 | 0.78 | 0.63 | 0.86 |





**Table 6: Exponential relationships between NDII values and simulated root zone moisture storage (*Su*) in six sub-basins controlled by runoff stations. Best performance in bold.**

| Station | Model | Dry season | | | | Wet season | | | |
|---|---|---|---|---|---|---|---|---|---|
| | | *a* | *b* | $R^2$ | *NSE* | *a* | *b* | $R^2$ | *NSE* |
| P.20 | FLEXL | 65.1 | 5.7 | 0.83 | 0.66 | 37.0 | 7.9 | **0.48** | 0.51 |
| | FLEX-SD | 25.6 | 9.0 | 0.79 | 0.60 | 14.0 | 11.7 | 0.43 | 0.52 |
| | FLEX-SD-NDII$_{Max-Min}$ | 133.8 | 4.5 | 0.81 | 0.60 | 81.6 | 6.2 | **0.48** | 0.46 |
| | FLEX-SD-NDII$_{Avg}$ | 42.1 | 7.3 | **0.84** | **0.67** | 23.2 | 9.9 | 0.47 | **0.54** |
| P.75 | FLEXL | 208.2 | 3.9 | 0.79 | 0.47 | 126.1 | 5.4 | 0.44 | 0.38 |
| | FLEX-SD | 23.3 | 9.5 | 0.80 | 0.62 | 12.0 | 12.4 | 0.43 | 0.53 |
| | FLEX-SD-NDII$_{Max-Min}$ | 123.1 | 4.9 | 0.82 | 0.61 | 71.6 | 6.7 | **0.48** | 0.46 |
| | FLEX-SD-NDII$_{Avg}$ | 44.1 | 7.5 | **0.84** | **0.68** | 23.1 | 10.1 | 0.47 | **0.54** |
| P.4A | FLEXL | 22.0 | 9.2 | 0.69 | 0.40 | 12.6 | 11.9 | 0.38 | 0.40 |
| | FLEX-SD | 6.4 | 12.8 | 0.69 | 0.30 | 5.3 | 15.0 | 0.36 | 0.41 |
| | FLEX-SD-NDII$_{Max-Min}$ | 36.7 | 7.8 | **0.74** | **0.45** | 25.0 | 10.0 | **0.41** | **0.44** |
| | FLEX-SD-NDII$_{Avg}$ | 51.7 | 7.0 | 0.71 | 0.43 | 32.2 | 9.2 | **0.41** | 0.42 |
| P.67 | FLEXL | 16.5 | 10.2 | 0.79 | 0.56 | 6.6 | 14.3 | 0.43 | 0.51 |
| | FLEX-SD | 13.9 | 11.0 | 0.79 | 0.56 | 7.1 | 14.4 | 0.45 | **0.54** |
| | FLEX-SD-NDII$_{Max-Min}$ | 72.6 | 6.3 | **0.82** | **0.63** | 39.5 | 8.7 | **0.51** | 0.53 |
| | FLEX-SD-NDII$_{Avg}$ | 51.1 | 7.0 | **0.82** | 0.62 | 26.5 | 9.8 | 0.49 | 0.53 |
| P.21 | FLEXL | 71.9 | 5.9 | 0.64 | 0.33 | 39.6 | 7.5 | 0.38 | 0.29 |
| | FLEX-SD | 8.5 | 12.4 | 0.69 | **0.54** | 4.7 | 14.5 | 0.35 | **0.40** |
| | FLEX-SD-NDII$_{Max-Min}$ | 56.1 | 7.2 | **0.70** | 0.49 | 30.7 | 8.9 | **0.39** | 0.36 |
| | FLEX-SD-NDII$_{Avg}$ | 52.9 | 7.2 | 0.68 | 0.45 | 28.5 | 8.9 | 0.38 | 0.34 |
| P.1 | FLEXL | 11.7 | 11.5 | 0.79 | 0.57 | 4.8 | 15.6 | 0.44 | 0.53 |
| | FLEX-SD | 13.1 | 11.3 | 0.78 | 0.58 | 6.4 | 14.8 | 0.46 | **0.54** |
| | FLEX-SD-NDII$_{Max-Min}$ | 69.4 | 6.5 | **0.81** | **0.63** | 36.3 | 9.1 | **0.51** | 0.53 |
| | FLEX-SD-NDII$_{Avg}$ | 49.8 | 7.2 | **0.81** | 0.62 | 24.8 | 10.1 | 0.50 | 0.52 |
| Average | FLEXL | - | - | 0.75 | 0.50 | - | - | 0.43 | 0.44 |
| | FLEX-SD | - | - | 0.76 | 0.53 | - | - | 0.41 | **0.49** |
| | FLEX-SD-NDII$_{Max-Min}$ | - | - | **0.79** | 0.57 | - | - | **0.46** | 0.46 |
| | FLEX-SD-NDII$_{Avg}$ | - | - | 0.78 | **0.58** | - | - | **0.46** | 0.48 |

Note: $Su = ae^{b(NDII)}$

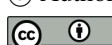



**Table 7:** Exponential relationships between the NDII values and simulated root zone moisture storage ($Su$) in 31 ungauged sub-basins. Best performance in bold.

| Model | Dry season | | Wet season | |
|---|---|---|---|---|
| | $R^2$ | NSE | $R^2$ | NSE |
| FLEX-SD | 0.71 | 0.41 | 0.36 | 0.38 |
| FLEX-SD-NDII$_{Max-Min}$ | **0.74** | 0.45 | **0.41** | 0.36 |
| FLEX-SD-NDII$_{Avg}$ | **0.74** | **0.52** | **0.41** | **0.40** |

**Table 8: Exponential relationships between the daily SWI040 values and simulated root zone moisture storage ($Su$) in six sub-basins controlled by runoff stations. Best performance in bold.**

| Station | Model | Dry season | | | | Wet season | | | |
|---|---|---|---|---|---|---|---|---|---|
| | | $a$ | $b$ | $R^2$ | NSE | $a$ | $b$ | $R^2$ | NSE |
| P.20 | FLEXL | 55.4 | 0.024 | 0.88 | 0.84 | 36.3 | 0.030 | **0.91** | 0.88 |
| | FLEX-SD | 20.2 | 0.037 | 0.87 | 0.79 | 15.1 | 0.042 | 0.88 | 0.86 |
| | FLEX-SD-NDII$_{Max-Min}$ | 118.7 | 0.019 | 0.85 | 0.78 | 78.8 | 0.024 | 0.90 | 0.84 |
| | FLEX-SD-NDII$_{Avg}$ | 34.4 | 0.031 | **0.91** | **0.85** | 24.0 | 0.037 | **0.91** | **0.89** |
| P.75 | FLEXL | 190.6 | 0.016 | 0.85 | 0.75 | 126.2 | 0.021 | 0.90 | 0.82 |
| | FLEX-SD | 20.9 | 0.037 | 0.89 | 0.81 | 15.6 | 0.043 | 0.89 | 0.87 |
| | FLEX-SD-NDII$_{Max-Min}$ | 116.1 | 0.019 | 0.87 | 0.81 | 76.5 | 0.025 | 0.91 | 0.85 |
| | FLEX-SD-NDII$_{Avg}$ | 40.3 | 0.029 | **0.92** | **0.86** | 27.4 | 0.036 | **0.92** | **0.89** |
| P.4A | FLEXL | 47.9 | 0.026 | 0.85 | 0.78 | 26.6 | 0.033 | 0.86 | 0.77 |
| | FLEX-SD | 19.7 | 0.036 | **0.89** | 0.81 | 14.8 | 0.040 | **0.89** | **0.83** |
| | FLEX-SD-NDII$_{Max-Min}$ | 71.8 | 0.022 | **0.89** | **0.84** | 49.3 | 0.027 | **0.89** | **0.83** |
| | FLEX-SD-NDII$_{Avg}$ | 92.9 | 0.020 | 0.84 | 0.76 | 59.5 | 0.025 | 0.85 | 0.77 |
| P.67 | FLEXL | 22.5 | 0.035 | **0.91** | 0.82 | 12.6 | 0.043 | **0.91** | 0.84 |
| | FLEX-SD | 20.1 | 0.037 | 0.90 | 0.81 | 15.4 | 0.042 | 0.90 | **0.86** |
| | FLEX-SD-NDII$_{Max-Min}$ | 88.8 | 0.021 | 0.89 | **0.84** | 60.2 | 0.026 | **0.91** | **0.86** |
| | FLEX-SD-NDII$_{Avg}$ | 63.4 | 0.024 | 0.89 | **0.84** | 41.9 | 0.030 | **0.91** | 0.85 |
| P.21 | FLEXL | 97.4 | 0.019 | 0.74 | 0.57 | 55.3 | 0.025 | 0.77 | 0.66 |
| | FLEX-SD | 16.8 | 0.039 | **0.87** | **0.73** | 10.2 | 0.047 | **0.83** | **0.77** |
| | FLEX-SD-NDII$_{Max-Min}$ | 82.1 | 0.023 | **0.84** | **0.73** | 47.5 | 0.030 | 0.82 | 0.74 |
| | FLEX-SD-NDII$_{Avg}$ | 77.3 | 0.023 | 0.81 | 0.69 | 43.6 | 0.030 | 0.80 | 0.71 |
| P.1 | FLEXL | 17.1 | 0.038 | **0.92** | 0.81 | 10.6 | 0.046 | **0.90** | **0.85** |
| | FLEX-SD | 19.5 | 0.037 | 0.90 | 0.80 | 14.6 | 0.043 | **0.90** | **0.85** |
| | FLEX-SD-NDII$_{Max-Min}$ | 86.5 | 0.022 | 0.89 | **0.83** | 57.4 | 0.027 | **0.90** | **0.85** |
| | FLEX-SD-NDII$_{Avg}$ | 63.1 | 0.024 | 0.89 | **0.83** | 40.8 | 0.030 | **0.90** | 0.84 |
| Average | FLEXL | - | - | 0.86 | 0.76 | - | - | 0.87 | 0.80 |
| | FLEX-SD | - | - | **0.89** | 0.79 | - | - | 0.88 | **0.84** |
| | FLEX-SD-NDII$_{Max-Min}$ | - | - | 0.87 | **0.81** | - | - | **0.89** | 0.83 |
| | FLEX-SD-NDII$_{Avg}$ | - | - | 0.88 | **0.81** | - | - | 0.88 | 0.83 |

Note: $Su = ae^{b(SWI)}$





15 **Table 9:** **Exponential relationships between the daily SWI040 values and simulated root zone moisture storage ($Su$) in 31 ungauged sub-basins for semi-distributed models. Best performance in bold.**

| Model | Dry season | | Wet season | |
|---|---|---|---|---|
| | $R^2$ | NSE | $R^2$ | NSE |
| FLEX-SD | **0.87** | 0.78 | **0.87** | **0.81** |
| FLEX-SD-NDII$_{Max-Min}$ | 0.86 | 0.78 | **0.87** | 0.79 |
| FLEX-SD-NDII$_{Avg}$ | **0.87** | **0.79** | **0.87** | 0.79 |

(a) Topography for each sub-catchment of the UPRB

(b) Rainfall depth for each sub-catchment of the UPRB

**Figure 1: Topography and mean annual rainfall depth for each sub-catchment of the UPRB**


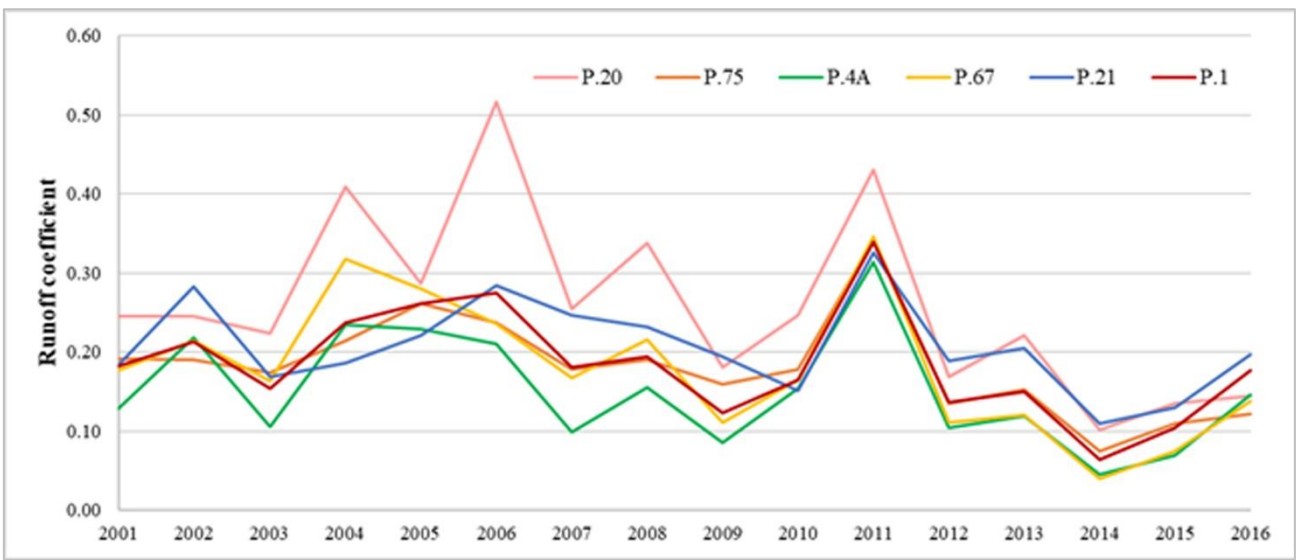

25    **Figure 2: Runoff coefficient of each station**

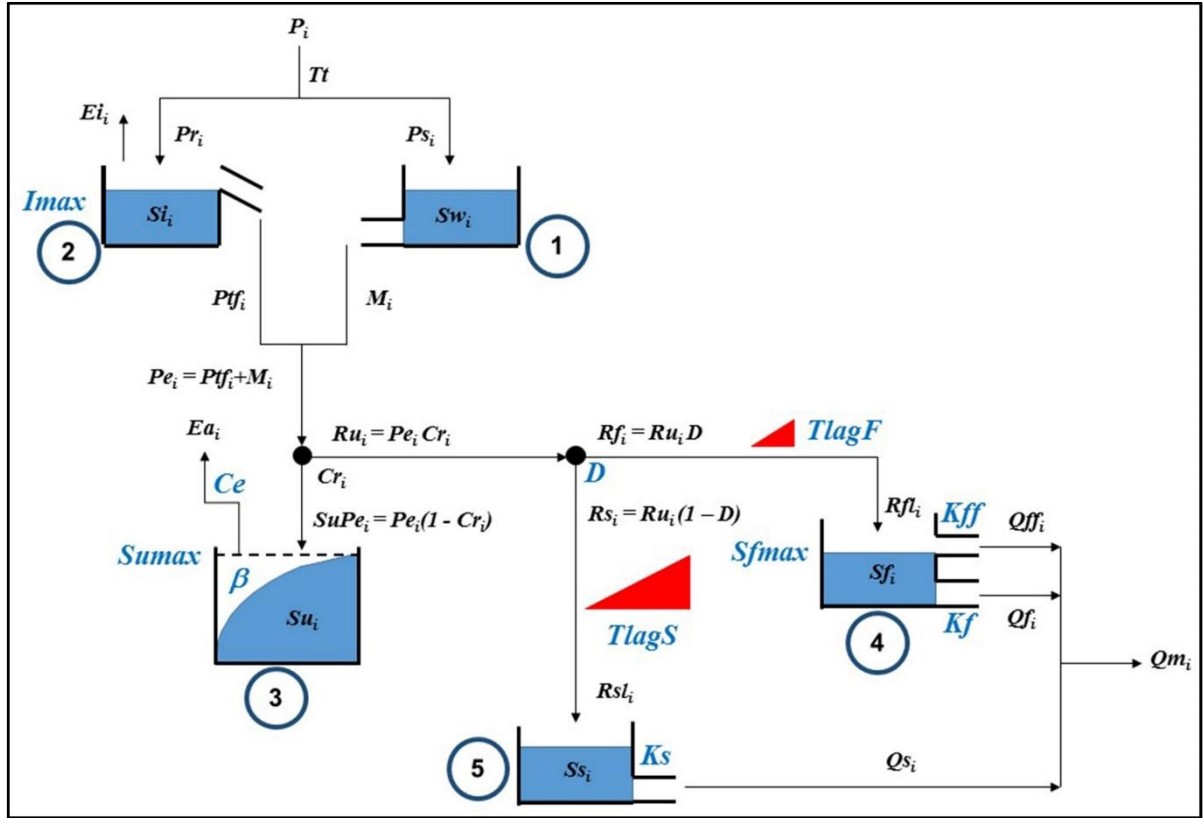

**Figure 3: Model structure of FLEXL model**







**Figure 4:** **Accumulated simulated and observed runoff at all stations produced by FLEXL calibration at each station and by semi-distributed model calibration at P.1**







**Figure 5:** **Hydrograph of simulated and observed runoff at 6 stations produced by FLEXL calibrated at each station and by semi-distributed model calibrated only at P.1**







**Figure 6:** **Flow duration curves of simulated and observed runoff at 6 stations produced by FLEXL calibrated at each station and by semi-distributed model calibrated only at P.1**






**Table A1: Model parameters of FLEXL (calibrated at all stations) and FLEX-SD and FLEX-SD-NDII (calibrated only at P.1)**

| Station | Model - Case | Imax (mm) | Sumax (mm) | Ce | β | D | Kf | Ks | TlagF (hr) | TlagS (hr) | Sfmax (mm) | Kff | α | X | b | R |
|---|---|---|---|---|---|---|---|---|---|---|---|---|---|---|---|---|
| **P.1** | (1) FLEXL | 1.59 | 475.80 | 0.93 | 0.22 | 0.69 | 37.87 | 111.42 | 3.33 | 20.07 | 3.22 | 6.85 | | | | |
| | (2) FLEX-SD | 3.51 | 435.48 | 0.69 | 0.48 | 0.82 | 8.12 | 36.68 | 5.03 | 56.42 | 8.63 | 3.47 | 0.30 | 0.19 | | |
| | (3) FLEX-SD-NDII_Max-Min | 2.22 | 476.49 | 0.96 | 0.26 | 0.72 | 13.27 | 16.58 | 3.58 | 79.46 | 7.46 | 4.30 | 0.22 | 0.10 | 12.76 | |
| | (4) FLEX-SD-NDII_Avg | 3.18 | 464.87 | 0.95 | 0.31 | 0.62 | 4.53 | 19.90 | 5.35 | 22.53 | 2.81 | 3.50 | 0.38 | 0.14 | 15.50 | 0.49 |
| **P.20** | (1) FLEXL | 2.85 | 411.45 | 0.89 | 0.68 | 0.72 | 6.37 | 41.52 | 2.64 | 73.69 | 14.12 | 3.09 | | | | |
| | (2) FLEX-SD | * | * | * | * | * | * | * | 3.71 | 41.62 | * | * | * | * | | |
| | (3) FLEX-SD-NDII_Max-Min | * | 599.76 | * | * | * | * | * | 2.64 | 58.61 | * | * | * | * | * | |
| | (4) FLEX-SD-NDII_Avg | * | 380.45 | * | * | * | * | * | 3.94 | 16.62 | * | * | * | * | * | * |
| **P.75** | (1) FLEXL | 1.98 | 514.21 | 0.86 | 0.30 | 0.55 | 11.08 | 165.45 | 4.09 | 15.35 | 1.11 | 8.00 | | | | |
| | (2) FLEX-SD | * | * | * | * | * | * | * | 6.44 | 72.19 | * | * | * | * | | |
| | (3) FLEX-SD-NDII_Max-Min | * | 462.51 | * | * | * | * | * | 4.58 | 101.66 | * | * | * | * | * | |
| | (4) FLEX-SD-NDII_Avg | * | 409.31 | * | * | * | * | * | 6.84 | 28.82 | * | * | * | * | * | * |
| **P.4A** | (1) FLEXL | 4.19 | 429.49 | 0.86 | 0.38 | 0.91 | 13.34 | 43.48 | 4.29 | 30.12 | 8.13 | 7.27 | | | | |
| | (2) FLEX-SD | * | * | * | * | * | * | * | 3.71 | 41.56 | * | * | * | * | | |
| | (3) FLEX-SD-NDII_Max-Min | * | 483.50 | * | * | * | * | * | 2.64 | 58.53 | * | * | * | * | * | |
| | (4) FLEX-SD-NDII_Avg | * | 563.47 | * | * | * | * | * | 3.94 | 16.60 | * | * | * | * | * | * |
| **P.67** | (1) FLEXL | 3.53 | 358.74 | 0.75 | 0.41 | 0.76 | 16.30 | 175.56 | 3.03 | 51.90 | 8.52 | 7.38 | | | | |
| | (2) FLEX-SD | * | * | * | * | * | * | * | 5.26 | 59.03 | * | * | * | * | | |
| | (3) FLEX-SD-NDII_Max-Min | * | 469.08 | * | * | * | * | * | 3.75 | 83.13 | * | * | * | * | * | |
| | (4) FLEX-SD-NDII_Avg | * | 460.79 | * | * | * | * | * | 5.59 | 23.57 | * | * | * | * | * | * |
| **P.21** | (1) FLEXL | 4.88 | 759.96 | 0.88 | 1.14 | 0.70 | 11.71 | 42.09 | 2.48 | 23.98 | 9.40 | 4.77 | | | | |
| | (2) FLEX-SD | * | * | * | * | * | * | * | 3.79 | 42.50 | * | * | * | * | | |
| | (3) FLEX-SD-NDII_Max-Min | * | 547.29 | * | * | * | * | * | 2.70 | 59.85 | * | * | * | * | * | |
| | (4) FLEX-SD-NDII_Avg | * | 543.68 | * | * | * | * | * | 4.03 | 16.97 | * | * | * | * | * | * |

Note: * Same parameter values as P.1 for FLEX-SD and FLEX-SD-NDII





**Table A2: Exponential relationships between the average NDII values and simulated root zone moisture storage ($Su$) in 31 sub-basins. Best performance in bold.**

| Station | Model | Dry season | | | | Wet season | | | |
|---|---|---|---|---|---|---|---|---|---|
| | | *a* | *b* | *R²* | *NSE* | *a* | *b* | *R²* | *NSE* |
| | FLEX-SD | 22.1 | 9.5 | 0.74 | 0.54 | 15.5 | 11.5 | 0.34 | 0.49 |
| Sub 2 | FLEX-SD-NDII$_{Max-Min}$ | 56.5 | 6.8 | **0.81** | **0.65** | 37.7 | 8.7 | **0.42** | 0.52 |
| | FLEX-SD-NDII$_{Avg}$ | 35.9 | 8.0 | 0.80 | 0.63 | 24.2 | 9.9 | 0.39 | **0.53** |
| | FLEX-SD | 19.4 | 10.0 | 0.72 | 0.51 | 14.1 | 12.1 | 0.33 | 0.50 |
| Sub 3 | FLEX-SD-NDII$_{Max-Min}$ | 170.6 | 4.1 | 0.71 | 0.48 | 119.2 | 5.3 | **0.44** | 0.43 |
| | FLEX-SD-NDII$_{Avg}$ | 31.4 | 8.5 | **0.79** | **0.60** | 21.9 | 10.5 | 0.39 | **0.54** |
| | FLEX-SD | 57.1 | 6.9 | 0.77 | 0.65 | 31.4 | 8.9 | 0.31 | 0.48 |
| Sub 4 | FLEX-SD-NDII$_{Max-Min}$ | 364.2 | 2.4 | 0.69 | 0.15 | 242.6 | 3.1 | 0.34 | 0.17 |
| | FLEX-SD-NDII$_{Avg}$ | 39.3 | 7.6 | **0.83** | **0.73** | 24.9 | 9.5 | **0.35** | **0.62** |
| | FLEX-SD | 24.8 | 9.5 | 0.75 | 0.59 | 14.4 | 12.2 | 0.37 | 0.53 |
| Sub 5 | FLEX-SD-NDII$_{Max-Min}$ | 203.8 | 3.9 | 0.74 | 0.44 | 129.5 | 5.2 | **0.45** | 0.39 |
| | FLEX-SD-NDII$_{Avg}$ | 39.2 | 7.8 | **0.81** | **0.66** | 23.1 | 10.4 | 0.44 | **0.57** |
| | FLEX-SD | 24.7 | 9.4 | 0.77 | 0.60 | 13.8 | 12.2 | 0.43 | 0.54 |
| Sub 6 | FLEX-SD-NDII$_{Max-Min}$ | 162.5 | 4.2 | 0.78 | 0.54 | 101.5 | 5.8 | **0.49** | 0.45 |
| | FLEX-SD-NDII$_{Avg}$ | 39.4 | 7.7 | **0.83** | **0.67** | 22.3 | 10.4 | 0.48 | **0.58** |
| | FLEX-SD | 25.6 | 8.0 | 0.77 | 0.55 | 15.7 | 9.9 | 0.24 | **0.36** |
| Sub 7 | FLEX-SD-NDII$_{Max-Min}$ | 64.5 | 5.8 | **0.82** | **0.63** | 40.6 | 7.2 | **0.29** | **0.36** |
| | FLEX-SD-NDII$_{Avg}$ | 47.1 | 6.4 | 0.81 | 0.61 | 28.6 | 8.1 | 0.27 | **0.36** |
| | FLEX-SD | 29.5 | 7.6 | 0.78 | 0.55 | 17.3 | 9.7 | 0.28 | **0.41** |
| Sub 8 | FLEX-SD-NDII$_{Max-Min}$ | 71.1 | 5.4 | **0.83** | **0.64** | 43.4 | 7.1 | **0.34** | **0.41** |
| | FLEX-SD-NDII$_{Avg}$ | 48.3 | 6.2 | 0.82 | 0.62 | 28.5 | 8.2 | 0.32 | **0.41** |
| | FLEX-SD | 25.6 | 9.0 | 0.79 | 0.60 | 14.0 | 11.7 | 0.43 | 0.52 |
| Sub 9 | FLEX-SD-NDII$_{Max-Min}$ | 133.8 | 4.5 | 0.81 | 0.60 | 81.6 | 6.2 | **0.48** | 0.46 |
| | FLEX-SD-NDII$_{Avg}$ | 42.1 | 7.3 | **0.84** | **0.67** | 23.2 | 9.9 | 0.47 | **0.54** |
| | FLEX-SD | 24.4 | 9.2 | 0.79 | 0.61 | 13.2 | 12.0 | 0.43 | 0.52 |
| Sub 10 | FLEX-SD-NDII$_{Max-Min}$ | 125.6 | 4.7 | 0.81 | 0.61 | 75.4 | 6.5 | **0.48** | 0.47 |
| | FLEX-SD-NDII$_{Avg}$ | 42.5 | 7.4 | **0.83** | **0.67** | 23.1 | 10.0 | 0.47 | **0.54** |
| | FLEX-SD | 23.5 | 9.3 | 0.79 | 0.60 | 12.4 | 12.2 | 0.43 | 0.52 |
| Sub 11 | FLEX-SD-NDII$_{Max-Min}$ | 121.0 | 4.8 | 0.82 | 0.61 | 71.4 | 6.7 | **0.48** | 0.47 |
| | FLEX-SD-NDII$_{Avg}$ | 45.4 | 7.2 | **0.83** | **0.66** | 24.2 | 9.9 | 0.47 | **0.53** |
| | FLEX-SD | 23.3 | 9.5 | 0.80 | 0.62 | 12.0 | 12.4 | 0.43 | 0.53 |
| Sub 12 | FLEX-SD-NDII$_{Max-Min}$ | 123.1 | 4.9 | 0.82 | 0.61 | 71.6 | 6.7 | **0.48** | 0.46 |
| | FLEX-SD-NDII$_{Avg}$ | 44.1 | 7.5 | **0.84** | **0.68** | 23.1 | 10.1 | 0.47 | **0.54** |
| | FLEX-SD | 24.4 | 9.4 | 0.81 | 0.63 | 12.2 | 12.4 | 0.45 | 0.54 |
| Sub 13 | FLEX-SD-NDII$_{Max-Min}$ | 123.4 | 4.9 | 0.83 | 0.61 | 70.4 | 6.8 | **0.50** | 0.47 |
| | FLEX-SD-NDII$_{Avg}$ | 44.2 | 7.4 | **0.85** | **0.69** | 22.7 | 10.1 | **0.50** | **0.55** |

Note: $Su = ae^{b(NDII)}$





| Station | Model | Dry season | | | | Wet season | | | |
|---------|-------|:---:|:---:|:---:|:---:|:---:|:---:|:---:|:---:|
| | | *a* | *b* | *R²* | *NSE* | *a* | *b* | *R²* | *NSE* |
| Sub 14 | FLEX-SD | 2.8 | 12.7 | **0.34** | -1.12 | 73.9 | 2.7 | 0.05 | -0.74 |
| | FLEX-SD-NDII_Max-Min | 23.7 | 7.8 | **0.34** | -0.80 | 121.8 | 2.7 | 0.07 | -0.58 |
| | FLEX-SD-NDII_Avg | 73.6 | 5.3 | 0.30 | **-0.47** | 162.7 | 2.8 | **0.09** | **-0.41** |
| Sub 15 | FLEX-SD | 18.9 | 10.3 | 0.71 | 0.55 | 19.3 | 11.2 | 0.51 | 0.57 |
| | FLEX-SD-NDII_Max-Min | 9.3 | 11.9 | 0.76 | 0.48 | 15.3 | 11.6 | 0.50 | 0.60 |
| | FLEX-SD-NDII_Avg | 27.3 | 9.0 | **0.78** | **0.62** | 26.2 | 10.0 | **0.54** | **0.61** |
| Sub 16 | FLEX-SD | 17.4 | 10.9 | 0.75 | 0.59 | 17.2 | 11.8 | 0.52 | **-0.05** |
| | FLEX-SD-NDII_Max-Min | 14.3 | 11.0 | **0.81** | 0.58 | 18.1 | 11.4 | 0.53 | -0.10 |
| | FLEX-SD-NDII_Avg | 26.0 | 9.5 | **0.81** | **0.66** | 24.1 | 10.5 | **0.55** | -0.09 |
| Sub 17 | FLEX-SD | 9.5 | 12.4 | 0.73 | 0.47 | 10.2 | 13.5 | 0.45 | 0.52 |
| | FLEX-SD-NDII_Max-Min | 17.1 | 10.3 | **0.78** | 0.52 | 18.1 | 11.3 | 0.48 | **0.55** |
| | FLEX-SD-NDII_Avg | 36.3 | 8.2 | 0.77 | **0.58** | 30.0 | 9.7 | **0.51** | **0.55** |
| Sub 18 | FLEX-SD | 7.8 | 12.7 | 0.71 | 0.37 | 8.6 | 14.0 | 0.42 | 0.47 |
| | FLEX-SD-NDII_Max-Min | 18.7 | 9.8 | **0.76** | 0.45 | 18.4 | 11.2 | 0.46 | 0.50 |
| | FLEX-SD-NDII_Avg | 38.8 | 7.9 | 0.74 | **0.51** | 30.7 | 9.6 | **0.48** | **0.51** |
| Sub 19 | FLEX-SD | 7.2 | 12.8 | 0.70 | 0.32 | 7.7 | 14.3 | 0.40 | 0.45 |
| | FLEX-SD-NDII_Max-Min | 23.0 | 9.2 | **0.75** | 0.43 | 20.7 | 10.8 | 0.45 | **0.48** |
| | FLEX-SD-NDII_Avg | 39.7 | 7.8 | 0.72 | **0.47** | 30.1 | 9.6 | **0.46** | **0.48** |
| Sub 20 | FLEX-SD | 10.9 | 10.9 | 0.63 | 0.25 | 10.4 | 12.0 | 0.22 | **0.24** |
| | FLEX-SD-NDII_Max-Min | 90.0 | 5.5 | **0.66** | **0.36** | 68.3 | 6.6 | **0.27** | 0.23 |
| | FLEX-SD-NDII_Avg | 53.9 | 6.6 | **0.66** | 0.35 | 41.0 | 7.8 | 0.26 | 0.23 |
| Sub 21 | FLEX-SD | 7.6 | 12.5 | 0.69 | 0.31 | 6.9 | 14.5 | 0.38 | 0.44 |
| | FLEX-SD-NDII_Max-Min | 42.3 | 7.4 | **0.73** | **0.45** | 30.9 | 9.5 | **0.44** | **0.47** |
| | FLEX-SD-NDII_Avg | 42.3 | 7.5 | 0.71 | **0.45** | 28.8 | 9.7 | 0.43 | 0.46 |
| Sub 22 | FLEX-SD | 7.4 | 12.4 | 0.69 | 0.30 | 6.5 | 14.6 | 0.37 | 0.43 |
| | FLEX-SD-NDII_Max-Min | 39.8 | 7.5 | **0.74** | 0.44 | 28.4 | 9.7 | **0.43** | **0.46** |
| | FLEX-SD-NDII_Avg | 44.4 | 7.3 | 0.71 | **0.44** | 29.1 | 9.6 | 0.42 | 0.44 |
| Sub 23 | FLEX-SD | 2.5 | 13.8 | 0.45 | -0.31 | 18.6 | 7.8 | 0.10 | -0.33 |
| | FLEX-SD-NDII_Max-Min | 32.8 | 7.6 | **0.46** | -0.05 | 68.3 | 5.2 | 0.11 | -0.24 |
| | FLEX-SD-NDII_Avg | 88.1 | 5.4 | 0.41 | **0.05** | 117.8 | 4.2 | **0.13** | **-0.18** |
| Sub 24 | FLEX-SD | 6.0 | 12.9 | 0.67 | 0.24 | 5.4 | 14.9 | 0.35 | 0.38 |
| | FLEX-SD-NDII_Max-Min | 37.9 | 7.7 | **0.72** | **0.40** | 26.6 | 9.8 | **0.40** | **0.42** |
| | FLEX-SD-NDII_Avg | 52.9 | 6.9 | 0.68 | 0.39 | 33.9 | 9.0 | **0.40** | 0.40 |
| Sub 25 | FLEX-SD | 6.4 | 12.8 | 0.69 | 0.30 | 5.3 | 15.0 | 0.36 | 0.41 |
| | FLEX-SD-NDII_Max-Min | 36.7 | 7.8 | **0.74** | **0.45** | 25.0 | 10.0 | **0.41** | **0.44** |
| | FLEX-SD-NDII_Avg | 51.7 | 7.0 | 0.71 | 0.43 | 32.2 | 9.2 | **0.41** | 0.42 |
| Sub 26 | FLEX-SD | 13.9 | 11.0 | 0.79 | 0.56 | 7.1 | 14.4 | 0.45 | 0.54 |
| | FLEX-SD-NDII_Max-Min | 72.6 | 6.3 | **0.82** | **0.63** | 39.5 | 8.7 | **0.51** | **0.53** |
| | FLEX-SD-NDII_Avg | 51.1 | 7.0 | **0.82** | 0.62 | 26.5 | 9.8 | 0.49 | **0.53** |

Note: $Su = ae^{b(NDII)}$





| Station | Model | Dry season | | | | Wet season | | | |
|---|---|---|---|---|---|---|---|---|---|
| | | *a* | *b* | $R^2$ | *NSE* | *a* | *b* | $R^2$ | *NSE* |
| Sub 27 | FLEX-SD | 14.0 | 11.1 | 0.79 | 0.58 | 7.2 | 14.4 | 0.46 | **0.55** |
| | FLEX-SD-NDII$_{Max-Min}$ | 72.4 | 6.3 | **0.83** | **0.64** | 39.3 | 8.8 | **0.52** | 0.53 |
| | FLEX-SD-NDII$_{Avg}$ | 50.7 | 7.1 | 0.82 | 0.63 | 26.3 | 9.9 | 0.50 | 0.53 |
| Sub 28 | FLEX-SD | 14.9 | 10.0 | 0.76 | **0.57** | 7.9 | 12.0 | 0.38 | **0.42** |
| | FLEX-SD-NDII$_{Max-Min}$ | 67.7 | 6.1 | **0.77** | 0.54 | 37.7 | 7.6 | **0.41** | 0.38 |
| | FLEX-SD-NDII$_{Avg}$ | 54.4 | 6.6 | **0.77** | 0.53 | 29.0 | 8.2 | 0.40 | 0.37 |
| Sub 29 | FLEX-SD | 2.8 | 14.9 | **0.46** | 0.11 | 10.6 | 10.5 | 0.16 | -0.08 |
| | FLEX-SD-NDII$_{Max-Min}$ | 40.4 | 7.8 | **0.46** | **0.18** | 56.1 | 6.4 | **0.17** | **-0.06** |
| | FLEX-SD-NDII$_{Avg}$ | 54.8 | 7.0 | 0.42 | 0.13 | 65.2 | 6.0 | **0.17** | -0.08 |
| Sub 30 | FLEX-SD | 8.5 | 12.4 | 0.69 | **0.54** | 4.7 | 14.5 | 0.35 | **0.40** |
| | FLEX-SD-NDII$_{Max-Min}$ | 56.1 | 7.2 | **0.70** | 0.49 | 30.7 | 8.9 | **0.39** | 0.36 |
| | FLEX-SD-NDII$_{Avg}$ | 52.9 | 7.2 | 0.68 | 0.45 | 28.5 | 8.9 | 0.38 | 0.34 |
| Sub 31 | FLEX-SD | 13.2 | 11.3 | 0.78 | 0.58 | 6.5 | 14.7 | 0.46 | **0.54** |
| | FLEX-SD-NDII$_{Max-Min}$ | 70.8 | 6.4 | **0.81** | **0.63** | 37.1 | 9.0 | **0.51** | 0.52 |
| | FLEX-SD-NDII$_{Avg}$ | 50.7 | 7.2 | **0.81** | 0.62 | 25.4 | 10.0 | 0.50 | 0.52 |
| Sub 32 | FLEX-SD | 13.1 | 11.3 | 0.78 | 0.58 | 6.4 | 14.8 | 0.46 | **0.54** |
| | FLEX-SD-NDII$_{Max-Min}$ | 69.4 | 6.5 | **0.81** | **0.63** | 36.3 | 9.1 | **0.51** | 0.53 |
| | FLEX-SD-NDII$_{Avg}$ | 49.8 | 7.2 | **0.81** | 0.62 | 24.8 | 10.1 | 0.50 | 0.52 |
| Average | FLEX-SD | - | - | 0.71 | 0.41 | - | - | 0.36 | 0.38 |
| | FLEX-SD-NDII$_{Max-Min}$ | - | - | **0.74** | 0.45 | - | - | **0.41** | 0.36 |
| | FLEX-SD-NDII$_{Avg}$ | - | - | **0.74** | **0.52** | - | - | **0.41** | **0.40** |

Note: $Su = ae^{b(NDII)}$





**Table A3: Exponential relationships between the daily SWI040 values and simulated root zone moisture storage ($Su$) in 31 sub-basins. Best performance in bold.**

| Station | Model | Dry season | | | | Wet season | | | |
|---|---|---|---|---|---|---|---|---|---|
| | | $a$ | $b$ | $R^2$ | NSE | $a$ | $b$ | $R^2$ | NSE |
| Sub 2 | FLEX-SD | 23.4 | 0.036 | 0.85 | 0.76 | 18.4 | 0.041 | 0.86 | 0.84 |
| | FLEX-SD-NDII$_{Max-Min}$ | 58.1 | 0.026 | **0.91** | **0.86** | 41.8 | 0.031 | **0.90** | **0.89** |
| | FLEX-SD-NDII$_{Avg}$ | 37.1 | 0.030 | **0.91** | 0.84 | 27.6 | 0.035 | **0.90** | **0.89** |
| Sub 3 | FLEX-SD | 21.8 | 0.037 | 0.84 | 0.76 | 17.1 | 0.041 | 0.86 | 0.83 |
| | FLEX-SD-NDII$_{Max-Min}$ | 177.0 | 0.015 | 0.76 | 0.66 | 122.5 | 0.019 | 0.85 | 0.78 |
| | FLEX-SD-NDII$_{Avg}$ | 34.4 | 0.031 | **0.90** | **0.84** | 25.4 | 0.036 | **0.90** | **0.87** |
| Sub 4 | FLEX-SD | 21.7 | 0.037 | 0.86 | **0.77** | 14.7 | 0.044 | **0.85** | 0.85 |
| | FLEX-SD-NDII$_{Max-Min}$ | 259.9 | 0.013 | 0.72 | 0.50 | 168.2 | 0.018 | 0.81 | 0.70 |
| | FLEX-SD-NDII$_{Avg}$ | 13.7 | 0.040 | **0.93** | **0.77** | 12.5 | 0.044 | **0.85** | **0.90** |
| Sub 5 | FLEX-SD | 21.2 | 0.037 | 0.86 | 0.78 | 15.7 | 0.042 | 0.88 | 0.85 |
| | FLEX-SD-NDII$_{Max-Min}$ | 190.8 | 0.015 | 0.76 | 0.63 | 125.8 | 0.020 | 0.85 | 0.76 |
| | FLEX-SD-NDII$_{Avg}$ | 34.0 | 0.030 | **0.91** | **0.84** | 24.3 | 0.036 | **0.91** | **0.89** |
| Sub 6 | FLEX-SD | 20.8 | 0.037 | 0.86 | 0.79 | 15.7 | 0.042 | 0.88 | 0.85 |
| | FLEX-SD-NDII$_{Max-Min}$ | 150.6 | 0.017 | 0.81 | 0.71 | 100.8 | 0.021 | 0.88 | 0.81 |
| | FLEX-SD-NDII$_{Avg}$ | 33.8 | 0.031 | **0.91** | **0.85** | 24.2 | 0.036 | **0.91** | **0.89** |
| Sub 7 | FLEX-SD | 17.2 | 0.038 | 0.87 | 0.75 | 10.6 | 0.047 | 0.85 | 0.84 |
| | FLEX-SD-NDII$_{Max-Min}$ | 48.2 | 0.027 | **0.90** | **0.83** | 28.5 | 0.036 | **0.90** | **0.87** |
| | FLEX-SD-NDII$_{Avg}$ | 34.0 | 0.031 | 0.89 | 0.81 | 19.3 | 0.040 | 0.88 | 0.85 |
| Sub 8 | FLEX-SD | 18.0 | 0.038 | 0.87 | 0.77 | 11.7 | 0.046 | 0.86 | 0.85 |
| | FLEX-SD-NDII$_{Max-Min}$ | 49.9 | 0.027 | **0.90** | **0.84** | 30.7 | 0.035 | **0.90** | **0.88** |
| | FLEX-SD-NDII$_{Avg}$ | 32.2 | 0.032 | **0.90** | 0.82 | 19.5 | 0.040 | 0.88 | 0.87 |
| Sub 9 | FLEX-SD | 20.2 | 0.037 | 0.87 | 0.79 | 15.1 | 0.042 | 0.88 | 0.86 |
| | FLEX-SD-NDII$_{Max-Min}$ | 118.7 | 0.019 | 0.85 | 0.78 | 78.8 | 0.024 | 0.90 | 0.84 |
| | FLEX-SD-NDII$_{Avg}$ | 34.4 | 0.031 | **0.91** | **0.85** | 24.0 | 0.037 | **0.91** | **0.89** |
| Sub 10 | FLEX-SD | 20.5 | 0.037 | 0.88 | 0.80 | 15.3 | 0.042 | 0.88 | 0.87 |
| | FLEX-SD-NDII$_{Max-Min}$ | 114.9 | 0.019 | 0.85 | 0.79 | 76.3 | 0.024 | **0.91** | 0.85 |
| | FLEX-SD-NDII$_{Avg}$ | 36.8 | 0.030 | **0.91** | **0.85** | 25.3 | 0.036 | **0.91** | **0.89** |
| Sub 11 | FLEX-SD | 20.7 | 0.037 | 0.88 | 0.80 | 15.5 | 0.042 | 0.89 | 0.87 |
| | FLEX-SD-NDII$_{Max-Min}$ | 113.4 | 0.019 | 0.86 | 0.80 | 75.2 | 0.025 | 0.91 | 0.85 |
| | FLEX-SD-NDII$_{Avg}$ | 41.0 | 0.029 | **0.91** | **0.86** | 27.7 | 0.035 | **0.92** | **0.89** |
| Sub 12 | FLEX-SD | 20.9 | 0.037 | 0.89 | 0.81 | 15.6 | 0.043 | 0.89 | 0.87 |
| | FLEX-SD-NDII$_{Max-Min}$ | 116.1 | 0.019 | 0.87 | 0.81 | 76.5 | 0.025 | 0.91 | 0.85 |
| | FLEX-SD-NDII$_{Avg}$ | 40.3 | 0.029 | **0.92** | **0.86** | 27.4 | 0.036 | **0.92** | **0.89** |
| Sub 13 | FLEX-SD | 20.6 | 0.038 | 0.89 | 0.80 | 15.6 | 0.043 | 0.89 | 0.87 |
| | FLEX-SD-NDII$_{Max-Min}$ | 112.7 | 0.020 | 0.87 | 0.81 | 74.4 | 0.025 | 0.91 | 0.86 |
| | FLEX-SD-NDII$_{Avg}$ | 38.6 | 0.030 | **0.92** | **0.86** | 26.7 | 0.036 | **0.92** | **0.90** |

Note: $Su = ae^{b(SWI)}$





| Station | Model | Dry season | | | | Wet season | | | |
|---------|-------|------|------|-------|------|------|------|-------|------|
| | | *a* | *b* | $R^2$ | *NSE* | *a* | *b* | $R^2$ | *NSE* |
| Sub 14 | FLEX-SD | 19.1 | 0.035 | **0.89** | 0.80 | 17.7 | 0.037 | **0.87** | **0.82** |
| | FLEX-SD-NDII$_{Max-Min}$ | 76.0 | 0.022 | **0.89** | **0.85** | 56.2 | 0.025 | **0.87** | 0.81 |
| | FLEX-SD-NDII$_{Avg}$ | 158.9 | 0.015 | 0.78 | 0.66 | 110.7 | 0.018 | 0.79 | 0.69 |
| Sub 15 | FLEX-SD | 22.3 | 0.036 | 0.82 | 0.77 | 18.2 | 0.039 | 0.86 | 0.80 |
| | FLEX-SD-NDII$_{Max-Min}$ | 12.1 | 0.040 | **0.91** | 0.74 | 16.5 | 0.038 | 0.82 | **0.85** |
| | FLEX-SD-NDII$_{Avg}$ | 31.5 | 0.031 | 0.89 | **0.83** | 24.9 | 0.035 | **0.87** | 0.84 |
| Sub 16 | FLEX-SD | 22.7 | 0.035 | 0.85 | 0.79 | 19.6 | 0.038 | 0.86 | **0.04** |
| | FLEX-SD-NDII$_{Max-Min}$ | 19.7 | 0.035 | **0.91** | 0.79 | 22.1 | 0.035 | 0.85 | 0.00 |
| | FLEX-SD-NDII$_{Avg}$ | 33.0 | 0.031 | 0.89 | **0.84** | 27.0 | 0.034 | **0.87** | 0.01 |
| Sub 17 | FLEX-SD | 21.0 | 0.035 | 0.85 | 0.78 | 18.5 | 0.038 | 0.86 | 0.81 |
| | FLEX-SD-NDII$_{Max-Min}$ | 33.0 | 0.029 | **0.90** | **0.83** | 30.3 | 0.031 | **0.87** | **0.85** |
| | FLEX-SD-NDII$_{Avg}$ | 59.5 | 0.024 | 0.86 | 0.82 | 44.8 | 0.027 | 0.86 | 0.81 |
| Sub 18 | FLEX-SD | 20.6 | 0.035 | 0.85 | 0.78 | 17.8 | 0.038 | 0.86 | 0.81 |
| | FLEX-SD-NDII$_{Max-Min}$ | 39.3 | 0.027 | **0.89** | **0.83** | 33.3 | 0.031 | **0.87** | **0.85** |
| | FLEX-SD-NDII$_{Avg}$ | 68.5 | 0.022 | 0.84 | 0.79 | 49.4 | 0.026 | 0.85 | 0.79 |
| Sub 19 | FLEX-SD | 20.2 | 0.036 | 0.85 | 0.78 | 17.2 | 0.038 | 0.86 | 0.81 |
| | FLEX-SD-NDII$_{Max-Min}$ | 47.8 | 0.026 | **0.88** | **0.83** | 38.0 | 0.029 | **0.88** | **0.84** |
| | FLEX-SD-NDII$_{Avg}$ | 72.0 | 0.022 | 0.83 | 0.78 | 50.6 | 0.026 | 0.85 | 0.79 |
| Sub 20 | FLEX-SD | 20.6 | 0.036 | **0.88** | **0.81** | 13.3 | 0.042 | **0.88** | **0.82** |
| | FLEX-SD-NDII$_{Max-Min}$ | 121.3 | 0.019 | 0.84 | 0.74 | 73.8 | 0.024 | 0.85 | 0.76 |
| | FLEX-SD-NDII$_{Avg}$ | 77.6 | 0.022 | 0.85 | 0.77 | 44.5 | 0.029 | 0.86 | 0.77 |
| Sub 21 | FLEX-SD | 20.2 | 0.036 | **0.86** | 0.79 | 16.1 | 0.039 | **0.87** | **0.82** |
| | FLEX-SD-NDII$_{Max-Min}$ | 74.1 | 0.022 | **0.86** | **0.82** | 53.4 | 0.026 | **0.87** | **0.82** |
| | FLEX-SD-NDII$_{Avg}$ | 74.0 | 0.022 | 0.84 | 0.78 | 49.5 | 0.027 | 0.85 | 0.78 |
| Sub 22 | FLEX-SD | 19.9 | 0.036 | **0.87** | 0.79 | 15.6 | 0.039 | 0.87 | **0.82** |
| | FLEX-SD-NDII$_{Max-Min}$ | 71.0 | 0.022 | **0.87** | **0.83** | 50.5 | 0.026 | **0.88** | **0.82** |
| | FLEX-SD-NDII$_{Avg}$ | 77.1 | 0.022 | 0.83 | 0.78 | 50.6 | 0.026 | 0.85 | 0.78 |
| Sub 23 | FLEX-SD | 18.2 | 0.037 | **0.91** | **0.81** | 9.6 | 0.045 | **0.88** | **0.80** |
| | FLEX-SD-NDII$_{Max-Min}$ | 95.6 | 0.020 | 0.89 | 0.79 | 53.8 | 0.027 | 0.85 | 0.75 |
| | FLEX-SD-NDII$_{Avg}$ | 187.5 | 0.014 | 0.73 | 0.49 | 113.8 | 0.019 | 0.72 | 0.58 |
| Sub 24 | FLEX-SD | 19.7 | 0.036 | **0.88** | 0.80 | 14.8 | 0.040 | **0.88** | **0.83** |
| | FLEX-SD-NDII$_{Max-Min}$ | 75.8 | 0.022 | **0.88** | **0.83** | 51.5 | 0.026 | **0.88** | 0.82 |
| | FLEX-SD-NDII$_{Avg}$ | 96.7 | 0.020 | 0.83 | 0.75 | 61.9 | 0.024 | 0.84 | 0.76 |
| Sub 25 | FLEX-SD | 19.7 | 0.036 | **0.89** | 0.81 | 14.8 | 0.040 | **0.89** | **0.83** |
| | FLEX-SD-NDII$_{Max-Min}$ | 71.8 | 0.022 | **0.89** | **0.84** | 49.3 | 0.027 | **0.89** | **0.83** |
| | FLEX-SD-NDII$_{Avg}$ | 92.9 | 0.020 | 0.84 | 0.76 | 59.5 | 0.025 | 0.85 | 0.77 |
| Sub 26 | FLEX-SD | 20.1 | 0.037 | **0.90** | 0.81 | 15.4 | 0.042 | 0.90 | **0.86** |
| | FLEX-SD-NDII$_{Max-Min}$ | 88.8 | 0.021 | 0.89 | **0.84** | 60.2 | 0.026 | **0.91** | **0.86** |
| | FLEX-SD-NDII$_{Avg}$ | 63.4 | 0.024 | 0.89 | **0.84** | 41.9 | 0.030 | **0.91** | 0.85 |

Note: $Su = ae^{b(SWI)}$





**Table A3:** continued

| Station | Model | Dry season | | | | Wet season | | | |
|---------|-------|------------|---|---|---|------------|---|---|---|
| | | $a$ | $b$ | $R^2$ | NSE | $a$ | $b$ | $R^2$ | NSE |
| Sub 27 | FLEX-SD | 20.0 | 0.037 | **0.90** | 0.81 | 15.4 | 0.042 | 0.90 | **0.86** |
| | FLEX-SD-NDII<sub>Max-Min</sub> | 88.2 | 0.021 | 0.89 | **0.84** | 59.7 | 0.026 | **0.91** | **0.86** |
| | FLEX-SD-NDII<sub>Avg</sub> | 62.6 | 0.024 | **0.90** | **0.84** | 41.5 | 0.030 | **0.91** | 0.85 |
| Sub 28 | FLEX-SD | 16.2 | 0.039 | 0.88 | 0.73 | 9.3 | 0.048 | 0.84 | **0.78** |
| | FLEX-SD-NDII<sub>Max-Min</sub> | 70.8 | 0.024 | **0.86** | **0.77** | 40.0 | 0.031 | **0.85** | 0.77 |
| | FLEX-SD-NDII<sub>Avg</sub> | 57.0 | 0.025 | 0.85 | 0.75 | 30.3 | 0.034 | 0.84 | 0.75 |
| Sub 29 | FLEX-SD | 17.5 | 0.038 | **0.86** | **0.74** | 10.5 | 0.046 | **0.82** | **0.76** |
| | FLEX-SD-NDII<sub>Max-Min</sub> | 104.6 | 0.020 | 0.81 | 0.68 | 60.4 | 0.026 | 0.80 | 0.70 |
| | FLEX-SD-NDII<sub>Avg</sub> | 127.5 | 0.018 | 0.72 | 0.54 | 72.3 | 0.024 | 0.74 | 0.61 |
| Sub 30 | FLEX-SD | 16.8 | 0.039 | **0.87** | **0.73** | 10.2 | 0.047 | **0.83** | **0.77** |
| | FLEX-SD-NDII<sub>Max-Min</sub> | 82.1 | 0.023 | 0.84 | **0.73** | 47.5 | 0.030 | 0.82 | 0.74 |
| | FLEX-SD-NDII<sub>Avg</sub> | 77.3 | 0.023 | 0.81 | 0.69 | 43.6 | 0.030 | 0.80 | 0.71 |
| Sub 31 | FLEX-SD | 19.6 | 0.037 | **0.90** | 0.80 | 14.8 | 0.042 | **0.90** | **0.85** |
| | FLEX-SD-NDII<sub>Max-Min</sub> | 88.1 | 0.021 | 0.89 | **0.83** | 58.5 | 0.027 | **0.90** | **0.85** |
| | FLEX-SD-NDII<sub>Avg</sub> | 64.1 | 0.024 | 0.89 | **0.83** | 41.5 | 0.030 | **0.90** | 0.84 |
| Sub 32 | FLEX-SD | 19.5 | 0.037 | **0.90** | 0.80 | 14.6 | 0.043 | **0.90** | **0.85** |
| | FLEX-SD-NDII<sub>Max-Min</sub> | 86.5 | 0.022 | 0.89 | **0.83** | 57.4 | 0.027 | **0.90** | **0.85** |
| | FLEX-SD-NDII<sub>Avg</sub> | 63.1 | 0.024 | 0.89 | **0.83** | 40.8 | 0.030 | **0.90** | 0.84 |
| Average | FLEX-SD | - | - | **0.87** | 0.78 | - | - | **0.87** | **0.81** |
| | FLEX-SD-NDII<sub>Max-Min</sub> | - | - | 0.86 | 0.78 | - | - | **0.87** | 0.79 |
| | FLEX-SD-NDII<sub>Avg</sub> | - | - | **0.87** | **0.79** | - | - | **0.87** | 0.79 |

Note: $Su = ae^{b(SWI)}$





**Figure A1: Accumulated simulated and observed runoff at all stations produced by all models calibration and validation at each station**








**Figure A2: Hydrograph of simulated and observed runoff at all stations produced by all models calibration and validation at each station**






**Figure A3: Duration curves of simulated and observed runoff at all stations produced by all models calibration and validation at each station**



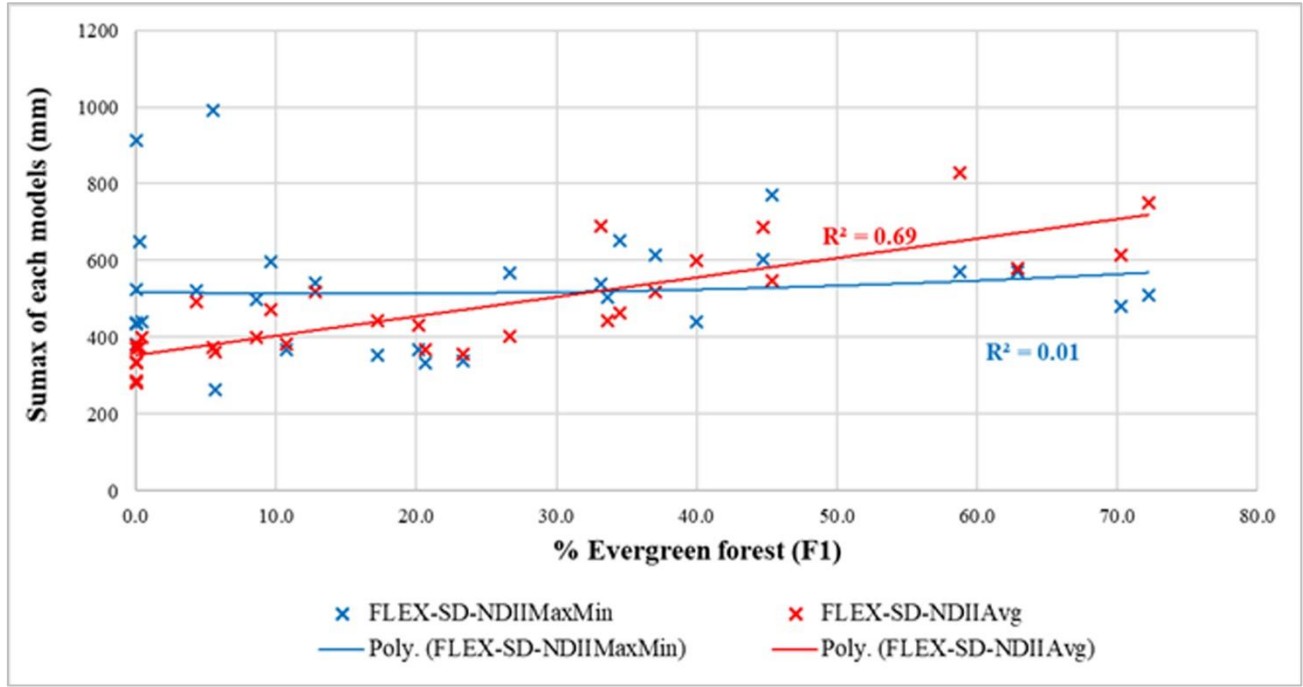

**Figure A4: Relationships between percent of evergreen forest and Sumax in 31 sub-catchments calibrated and validated by FLEX-SD-NDII$_{Avg}$ and FLEX-SD-NDII$_{MaxMins}$**

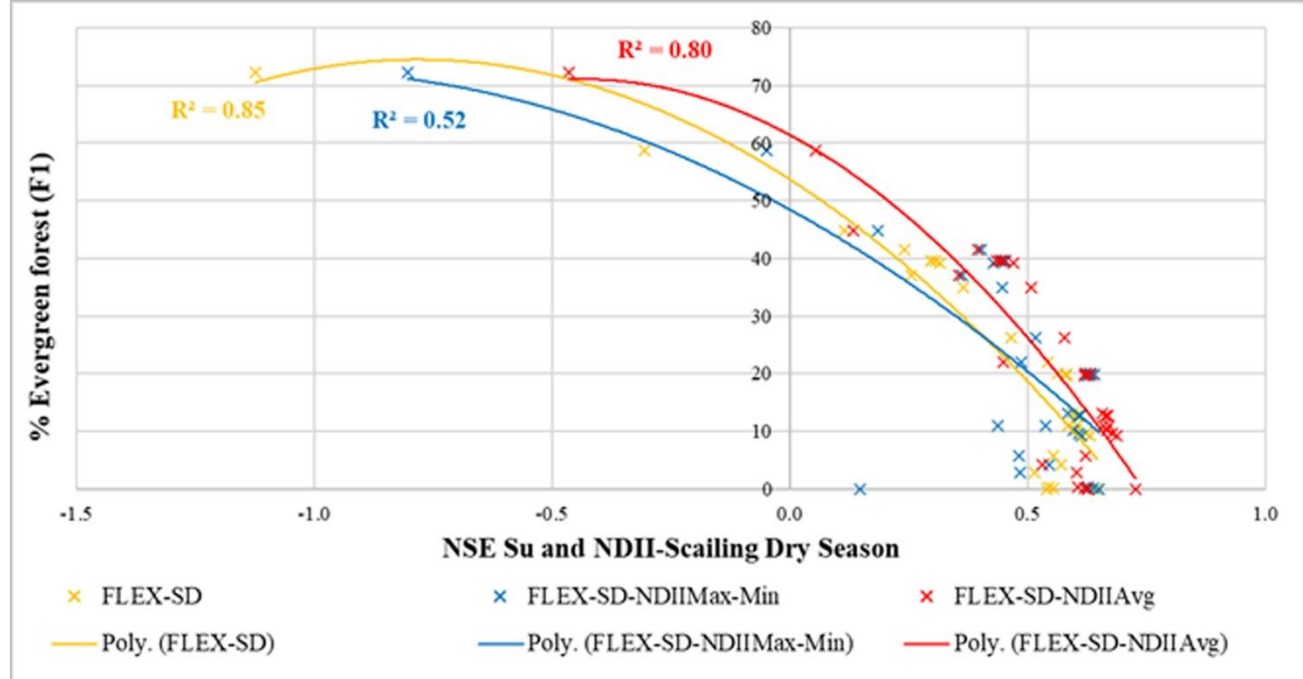

**Figure A5: Relationships between percent of evergreen forest and NSE values from the relationships between the average scaling NDII values and simulated root zone moisture storage (Su) in 31 sub-basins calibrated and validated by FLEX-SD, FLEX-SD-NDII$_{Avg}$ and FLEX-SD-NDII$_{Max-Min}$.**





**Figure A6:** Scatter plots between the average NDII and the average root zone moisture storage ($Su$) calculated with all models for
six runoff stations






**Figure A7: Scatter plots between the daily SWI and the daily root zone moisture storage (Sui) calculated with all models for six runoff stations**







**Figure A8: Time series plots of the average NDII (scaling), average SWI (scaling) and the average root zone moisture storage (Su) calculated with all models for six sub-basins controlled by runoff stations (dry season)**





**Figure A9: Time series plots of the average NDII (scaling), average SWI (scaling) and the average root zone moisture storage (Su) calculated with all models for six sub-basins controlled by runoff stations (wet season)**