# Peer review of "Using NDII patterns to constrain semi-distributed rainfall-runoff models in tropical nested catchments"

_Hydrology and Earth System Sciences, 2021_

## Community Comment (CC2)

The manuscript illustrates the calibration procedure of a semi-distributed rainfall-runoff model for flow prediction. The model is built based on the FLEXL model structure plus the Muskingum method for river routing, and it is applied to the Upper Ping catchment in Thailand. For this catchment there are 6 gauges and 32 sub-catchments are delineated. For flow prediction it is required to calibrate a large number of parameters for each sub-catchment; the calibration strategy relies on the observed discharge (between 2001 and 2016), the normalized difference infrared index (from 2002 to 2016) and the soil water Index for moisture conditions (from 2008 to 2016). Results are compared to those provided by a different modeling scheme, namely URBS model.

While the topic of this work is of interest for the scientific community, by providing additional developments for rainfall-runoff modeling, my general opinion is that the manuscript needs additional efforts from the Authors to be considered for publication in HESS. The main point here is that the research gaps motivating the present work and the innovative contribution to the literature are not clearly stated. As for results, a key issue is the estimation uncertainty, which should be quantified for model comparison in terms e.g. of prediction intervals. Due to the large number of calibration parameters, it is expected for the prediction intervals to be almost large. Hence, a fundamental aspect of this work should be how the information introduced here for calibration affects the prediction intervals (the estimation uncertainty) for different model structures. Note that only at the end of the manuscript (in the Conclusion Section) the Authors justify their work as a method to avoid uncertainty in runoff estimation (which is not avoidable in my opinion, but it can be reduced). Further, the text could be reorganized to be more concise and objective oriented, especially in the presentation of the methodology, yet not only. Finally, I suggest to revise figures to improve readability (e.g. remove "1 April" on x axis and use scientific notation in y-axis in figures 4 and A.1, increase text size in figures A.6-A.9).

*Answer:*

*We really appreciate the detailed and constructive review given by referee#1. We would like to answer his/her concern on estimation uncertainty as follows.*

FLEX-SD, FLEX-SD-NDII$_{Max-Min}$ and FLEX-SD-NDII$_{Avg}$ were calibrated (2001-2011) and validated (2012-2016) at P.1 station using 50,000 random parameter sets which were determined using the MOSCEM-UA algorithm by finding the Pareto-optimal solutions defined by three objective functions. These include the Kling-Gupta Efficiencies for high flows, low flows, and the flow duration (KGE$_E$, KGE$_L$ and KGE$_F$) respectively. To evaluate estimation uncertainty, the 5% best-performing parameter sets were identified as feasible (Hulsman et al., 2019) and were utilized to evaluate model performance. All around 2,500 parameter sets were used to create the box plots of KGE$_E$, KGE$_L$ and KGE$_F$ at the calibrated station (P.1) and at 5 upstream stations (P.20, P.4A, P.21, P.75 and P.67) (see **Figure 1**). The box plots provided by all models at P.21, P.75, P.67 and P.1 are similar, while FLEX-SD-NDII$_{Max-Min}$ performed slightly better than FLEX-SD and FLEX-SD-NDII$_{Avg}$. However, the box plots of KGE$_E$ and KGE$_F$ contributed by FLEX-SD-NDII$_{Avg}$ at P.4A and P.20 (tropical forest catchments) are exceptionally better than FLEX-SD and FLEX-SD-

NDII$_{Max-Min}$. Observed and calculated hydrographs acquired from the 5% best performing parameter combinations using FLEX-SD-NDII$_{Avg}$ at P.4A and P.20 show a narrow band compared to other 2 models but very similar at other stations, since all 3 KGE values of all models are similar, as shown in **Figure 2**.

In the revised paper, we shall present and discuss the model uncertainty, as required.

[Figure]

Figure 1: Comparison of box plots of the $KGE_E$, $KGE_L$ and $KGE_F$ at 6 gauging stations provided by 3 FLEX-SD models using 5% best-performing parameter sets

[Figure]

Figure 2: Comparison of the hydrographs at 6 gauging stations provided by 3 FLEX-SD models using 5% best-performing parameter sets

---

## Community Comment (CC5)

**Reply to anonymous Referee #2**

**Referee #2' s comment:**

The present manuscript describes an effort to incorporate the Normalized Difference Infrared Index (NDII) into a semi-distributed hydrological model based on the FLEXL model structure (including distributed time lags and channel routing routines), to drive partitioning of the water balance in favor of more realistic soil moisture storage capacities in the Upper Ping River basin in Thailand.

The effort to test remote sensing products readily available to the hydrological modelling community is scientifically relevant, and the proposals to obtain maximum moisture storage capacities from annual NDII values seem interesting and relatively novel to me. However, a broader bibliographic context on the work being done by other research groups worldwide seems to be missing from this article.

*Answer:*
*We thank you very much for your very careful review of our manuscript. We will utilize your comments and suggestions to improve the manuscript. Firstly, we will further look for research papers carried out by other groups related to our work. We would like to answer your other concerns and comments as follows.*

**Referee #2' s comment:**
It is my opinion that some of the conclusions presented in the paper are not supported by the modelling exercise, and some others are not substantial. After carefully reviewing the methodological strategy and the results, it seems to me that a part of the validation process is not rigorous enough from the point of view of causality. For example, when trying to validate with average NDII values the soil moisture contents that were simulated/constrained with the average and range of the very same NDII, as we would want to use independent (and ideally direct) observations to achieve rigorous validation. Following this precept, the use of the Soil Wetness Index (SWI) for validation purposes is more appropriate, although it is partly model-based and does not constitute either a direct observation or an indirect one that was empirically validated on direct measurements in the study area. Furthermore, the good correlation between the SWI and all models (whether NDII-based or not), both during the dry and wet seasons, leaves the impression that the contribution of the NDII is not fundamentally relevant to simulate the soil moisture component. It would be worth asking if the modelling could be made more efficient by constraining the soil moisture holding capacity routine rather with the SWI, which seems not to be particularly affected by the wet season effect, which was known in advance to be a limiting factor for the NDII.

*Answer:*
*We agree that it is not appropriate to validate the performance of the NDII-constrained models on the correspondence between computed Su and NDII. That is why we used the SWI as well. We will revise our manuscript by using only SWI for validation purpose. However, we do not think that SWI is a good constraint for distributed modelling, because it is partly model-based. It would lead*

*to circular reasoning. NDII, however, is an independent observation. Therefore, using NDII as a constraint and using SWI as a validity check is a better approach.*

*In the final paper we shall do a formal split-series calibration-validation test and uncertainty analysis (requested by referee#1) of the different models which shows the added value of using the NDII. That SWI works better during wet periods is not a real advantage. The soil moisture holding capacity is far stronger constrained by the stress periods, which is during the dry season. In the final check against the SWI we demonstrate that out approach also results in realistic values during wet periods and in wet ecosystems, such as evergreen forest.*

**Referee #2' s comment:**

Although in general the article seems well written to me, the wording of section 4.2 could be improved in order to avoid ambiguities (lines from 10 to 15 are not very straightforward). Tables 4 and 5 could also be improved, as they present duplicate data in some cases, and tabulations that do not seem appropriate. For example, by presenting results of the lumped FLEXL model calibrated for each of the various sub-catchments studied but tabulating these data under the table section: "for calibration at station P.1". Evidently a lumped model that was calibrated at P.1 could not provide results for individual upstream sub-catchments.

*Answer:*

*We will rewrite section 4.2 to avoid ambiguities as suggested by the referee. We will note in Tables 4 and 5 that FLEXL was calibrated at all gauging stations while all semi-distributed models were only calibrated at P.1.*

**Referee #2' s comments:**

I have trouble endorsing some interpretations and statements expressed in the results section. But I think the article is especially lacking in its conclusions, which I believe are not being formally supported and proven. For example, on p. 14, L. 16-17, it reads: "The results indicate that the relationship with the average root zone soil moisture storage is affected by the ecology of the river basin.". The term "ecology" is extremely broad and complex, and there is no definition and characterization of it in the article. Even less, a scientific demonstration of such soil moisture-ecology relationship. The article goes on to state that: "The results confirm the power of NDII to capture the spatial variation of root zone soil moisture within the sub-catchment scale", which, again, is neither accurate nor was it strictly proven through the scientific method.

*Answer:*

*We agree that the term ecology is out of place. The term land-use or land-cover (where appropriate) is more appropriate. We shall make this change in the re-submission. The sentence about the spatial variation should be reformulated as:* "The results confirm the power of NDII to represent spatial variation of root zone soil moisture between sub-catchments" *In this context it is good to observe that the average NDII value of each sub-catchment appears to be related with the percentages of evergreen forest of each sub-catchment.*

**Referee #2' s comment:**
On p. 15, L. 9-11, it reads: "As a result, the NDII appears to be useful to constrain hydrological models during dry conditions and both SWI and NDII appear to be useful to test model performance and to assess moisture states of river basins.". As stated earlier in this review, if NDII were used as a soil moisture model forcing, would not the simulation be expected to show this correlation? Furthermore, how sensitive is the model to forcing if even having constrained the soil moisture routine with the NDII, the result reveals a poor correlation between them during the wet season?

*Answer:*
*We agree that NDII should not be used to test model performance. We shall remove this comparison and only retain the comparison between Su and SWI. We also agree that NDII is not an appropriate indicator for non-moisture-constrained situations. This is well-known and also explained in the paper. The NDII is primarily a constraint on the moisture stress situation, which is more powerful to constrain the root zone moisture stress function. The comparison between the modeled Su and the SWI indicates that the models that were constrained on the NDII during moisture stress periods are quite capable of representing the wet periods as well.*

**Referee #2' s comment:**
On p. 15, L. 18-19, it reads: "… it has been shown that it is required to account for the spatial variation of the moisture holding capacity of the root zone.". In fact, this was not shown, considering that the best independent validation tool presented in the article is the SWI, which was shown to be highly correlated with the simulated soil moisture storage regardless of the inclusion of the NDII constrains in the model. In addition, including or not the spatial variation of the moisture holding capacity of the root zone is not necessarily required in all cases, but rather depends on the objectives of the modelling exercise. For example, if the objective is the best possible calibration of a rainfall-runoff model in the outlet of a basin for seasonal hydrological prediction purposes, as we have seen, in some cases a lumped model could generate greater efficiencies.

*Answer:*
*It is not surprising that a calibrated lumped model can generate more accurate runoff estimates at the outlet of the basin, but it requires model calibration at each station. This study proved that the developed semi-distributed rainfall-runoff models can provide as accurate runoff estimates at the calibrated station as well as at those stations upstream. This would make it possible to predict runoff at any require location within a river basin, without a gauging station being available. This is a great advantage for ungauged basins where flow and flood prediction is one of the main concerns of the hydrological community.*

**Referee #2' s comment:**
P. 15, L. 20-21: "We concluded that the maximum of a series of annual ranges (NDIIMaxMin) and annual average (NDIIAvg) of NDII values offers an effective proxy for estimating the appropriate Sumax values in the different sub-catchments. ". Again, it seems to me that the data presented cannot support this conclusion.

*Answer:*
*We will rewrite our conclusion properly in our revised manuscript.*

**Referee #2' s comment:**
P. 16, L. 1-2: "… However, during the wet season when soil moisture is replenished as a result of rainfall, NDII values are no longer well correlated with soil moisture.". In multiple parts of this article reference is made to "soil moisture" in order to later infer and conclude facts (such as its possible relationship with the NDII during the wet season), without the precaution that this soil moisture is not actually observed, but only a modeled value that was partially validated based on an indirect index (SWI). Due to the above, it does not seem appropriate to refer to soil moisture without specifying each time that it is a simulated value, nor drawing conclusions based on said simulated values, considering that they have not been sufficiently validated in the field and, therefore, cannot offer acceptable levels of accuracy and precision.

*Answer:*
*We will revise our manuscript accordingly.*

**Referee #2' s comment:**

In conclusion, I believe that this article could be substantially improved by better contextualizing it within the current global research environment on the specific topic of using remote sensing to improve soil moisture simulation in distributed hydrological models (how many other research initiatives around the word are trying to use the NDII to better simulate soil moisture storage capacity?). I would also suggest better organization, simplification and clarification of the description of the methodology (particularly section 4.2), and a thorough review of the modelling strategy, to avoid later interpretations that are irrelevant or based on spurious relationships (such as correlating an explicitly introduced forcing of the model outputs with those model outputs) or reaching conclusions that were not subject to hypothesis and testing (such as concluding about the ecology of a basin without having first systematized and analyzed that concept, or establishing the need to adopt a specific approach such as distributed modelling without sufficient empirical evidence to support it).

*Answer:*
*Thank you so much for all valuable comments and suggestions provided. We shall remove the comparison between Su and NDII and maintain the comparison with the SWI. We shall also include a formal calibration-validation on sub-basin runoff to demonstrate that the NDII-based method improved internal model performance.*

---

## Author Comment (AC3)

**Referee #1' s comments:**

The main point here is that the research gaps motivating the present work and the innovative contribution to the literature are not clearly stated. As for results, a key issue is the estimation uncertainty, which should be quantified for model comparison in terms e.g. of prediction intervals. Due to the large number of calibration parameters, it is expected for the prediction intervals to be almost large. Hence, a fundamental aspect of this work should be how the information introduced here for calibration affects the prediction intervals (the estimation uncertainty) for different model structures. Note that only at the end of the manuscript (in the Conclusion Section) the Authors justify their work as a method to avoid uncertainty in runoff estimation (which is not avoidable in my opinion, but it can be reduced).

While the topic of this work is of interest for the scientific community, by providing additional developments for rainfall-runoff modeling, my general opinion is that the manuscript needs additional efforts from the Authors to be considered for publication in HESS. The main point here is that the research gaps motivating the present work and the innovative contribution to the literature are not clearly stated. As for results, a key issue is the estimation uncertainty, which should be quantified for model comparison in terms e.g. of prediction intervals. Due to the large number of calibration parameters, it is expected for the prediction intervals to be almost large. Hence, a fundamental aspect of this work should be how the information introduced here for calibration affects the prediction intervals (the estimation uncertainty) for different model structures. Note that only at the end of the manuscript (in the Conclusion Section) the Authors justify their work as a method to avoid uncertainty in runoff estimation (which is not avoidable in my opinion, but it can be reduced).

*Answer:*

*Thank you very much for your close reading and comments. We appreciate your request to demonstrate if and how the different methods to constrain the parameter domain, by using spatial patterns of NDII, lead to less predictive uncertainty of the models. In this discussion forum, we demonstrate model uncertainty using the following procedures.*

*FLEX-SD, FLEX-SD-NDII$_{Max-Min}$ and FLEX-SD-NDII$_{Avg}$ were calibrated (2001-2011) and validated (2012-2016) at P.1 station using 50,000 random parameter sets which were determined using the MOSCEM-UA algorithm by finding the Pareto-optimal solutions defined by three objective functions. These include the Kling-Gupta Efficiencies for high flows, low flows, and the flow duration (KGEE, KGEL and KGEF) respectively. To evaluate estimation uncertainty, the 5% best-performing parameter sets were identified as feasible (Hulsman et al., 2019) and were utilized to evaluate model performance. All around 2,500 parameter sets were used to create the box plots of KGEE, KGEL and KGEF at the calibrated station (P.1) and at 5 upstream stations (P.20, P.4A, P.21, P.75 and P.67) (see Figure 1). The box plots provided by all models at P.21, P.75, P.67 and P.1 are similar, while FLEX-SD-NDII$_{Max-Min}$ performed slightly better than FLEX-SD and FLEX-SD-NDII$_{Avg}$. However, the box plots of KGEE and KGEF contributed by FLEX-SD-NDII$_{Avg}$ at P.4A and P.20 (tropical forest catchments) are exceptionally better than FLEX-SD and FLEX-SD-*

*$NDII_{Max-Min}$. Observed and calculated hydrographs acquired from the 5% best performing parameter combinations using FLEX-SD-NDII$_{Avg}$ at P.4A and P.20 show a narrow band compared to other 2 models but very similar at other stations, since all 3 KGE values of all models are similar, as shown in Figure 2.*

*In the revised paper, we shall present and discuss the model uncertainty, as required.*

*In the revised paper we shall describe better what the innovations of the paper are, because this has apparently not been brought out clear enough. We shall also take your other remarks at heart and revise the paper accordingly.*

**Referee #1' s comments:**

Further, the text could be reorganized to be more concise and objective oriented, especially in the presentation of the methodology, yet not only. Finally, I suggest to revise figures to improve readability (e.g. remove "1 April" on x axis and use scientific notation in y-axis in figures 4 and A.1, increase text size in figures A.6-A.9).

*Answer:*

*Thank you very much for the valued comments. We will revise our manuscript accordingly.*

[Figure]

***Figure 1***: *Comparison of box plots of the KGEE, KGEL and KGEF at 6 gauging stations provided by 3 FLEX-SD models using 5% best-performing parameter sets*

[Figure]

***Figure 2****: Comparison of the hydrographs at 6 gauging stations provided by 3 FLEX-SD models using 5% best-performing parameter sets*

---

## Author Response (AR1)

Dear Prof. Moussa,

Thank you again for giving us the opportunity to revise our paper. We understand very well that the use of RS-based NDII observations to assist distributed modelling is not a well-established approach and therefore has raised quite some questions by the reviewer. We have done our best to answer the observations and comments in detail and adjust the paper accordingly. We attach detailed replies to the reviewers and have indicated how we adjusted the paper. I think that in the process the paper has become much more clear and better substantiated. Also the graphs were substantially improved, which allowed us to remove a number of tables.

We trust that we have met the requirements and hope that this permits you to take a final decision.

On behalf of all my co-authors, sincerely,

Hubert Savenije

**Anonymous Referee #1**

**The referee #1' s comments:**

The manuscript illustrates the calibration procedure of a semi-distributed rainfall-runoff model for flow prediction. The model is built based on the FLEXL model structure plus the Muskingum method for river routing, and it is applied to the Upper Ping catchment in Thailand. For this catchment there are 6 gauges and 32 sub-catchments are delineated. For flow prediction it is required to calibrate a large number of parameters for each sub-catchment; the calibration strategy relies on the observed discharge (between 2001 and 2016), the normalized difference infrared index (from 2002 to 2016) and the soil water Index for moisture conditions (from 2008 to 2016). Results are compared to those provided by a different modeling scheme, namely URBS model.

While the topic of this work is of interest for the scientific community, by providing additional developments for rainfall-runoff modeling, my general opinion is that the manuscript needs additional efforts from the Authors to be considered for publication in HESS. The main point here is that the research gaps motivating the present work and the innovative contribution to the literature are not clearly stated. As for results, a key issue is the estimation uncertainty, which should be quantified for model comparison in terms e.g. of prediction intervals. Due to the large number of calibration parameters, it is expected for the prediction intervals to be almost large. Hence, a fundamental aspect of this work should be how the information introduced here for calibration affects the prediction intervals (the estimation uncertainty) for different model structures. Note that only at the end of the manuscript (in the Conclusion Section) the Authors

justify their work as a method to avoid uncertainty in runoff estimation (which is not avoidable in my opinion, but it can be reduced).

*Answer:*

*Thank you very much for your close reading and comments. We appreciate your request to demonstrate if and how the different methods to constrain the parameter domain, by using spatial patterns of NDII, lead to less predictive uncertainty of the models. In the revised manuscript, we demonstrate model uncertainty using the procedures described in 4.3: Estimation of Uncertainty in FLEX-SD-based models. The results are shown in 5.3: Uncertainty in runoff estimation using FLEX-SD, FLEX-SD-NDII$_{MaxMin}$ and FLEX-SD-NDII$_{Avg}$*

**The referee #1' s comments:**

Further, the text could be reorganized to be more concise and objective oriented, especially in the presentation of the methodology, yet not only. Finally, I suggest to revise figures to improve readability (e.g. remove "1 April" on x axis and use scientific notation in y-axis in figures 4 and A.1, increase text size in figures A.6-A.9).

*Answer:*

*In our revised manuscript, we have attempted to better emphasize the problem statement in the introduction section. Further, the revised methodology/results include sections on model uncertainties (4.3/5.3) and relationship between the average root zone soil moisture storage (Su$_i$) and the average NDII and SWI (4.4/5.4). The majority of tables were either replaced with figures to better present the tabulated results. Similarly, most figures have been reproduced to improve readability.*

**Reply to anonymous Referee #2**

**Referee #2' s comment:**

The present manuscript describes an effort to incorporate the Normalized Difference Infrared Index (NDII) into a semi-distributed hydrological model based on the FLEXL model structure (including distributed time lags and channel routing routines), to drive partitioning of the water balance in favor of more realistic soil moisture storage capacities in the Upper Ping River basin in Thailand.

The effort to test remote sensing products readily available to the hydrological modelling community is scientifically relevant, and the proposals to obtain maximum moisture storage capacities from annual NDII values seem interesting and relatively novel to me. However, a broader bibliographic context on the work being done by other research groups worldwide seems to be missing from this article.

**Answer:**

*As you have mentioned, the practice of incorporating remote sensing techniques (let alone NDII) is indeed novel. We have cited a few additional papers that have implemented the NDII product for monitoring plant water content and also drought. To our knowledge, papers which resemble our framework are not available.*

**Referee #2' s comment:**

It is my opinion that some of the conclusions presented in the paper are not supported by the modelling exercise, and some others are not substantial. After carefully reviewing the methodological strategy and the results, it seems to me that a part of the validation process is not rigorous enough from the point of view of causality. For example, when trying to validate with average NDII values the soil moisture contents that were simulated/constrained with the average and range of the very same NDII, as we would want to use independent (and ideally direct) observations to achieve rigorous validation. Following this precept, the use of the Soil Wetness Index (SWI) for validation purposes is more appropriate, although it is partly model-based and does not constitute either a direct observation or an indirect one that was empirically validated on direct measurements in the study area. Furthermore, the good correlation between the SWI and all models (whether NDII-based or not), both during the dry and wet seasons, leaves the impression that the contribution of the NDII is not fundamentally relevant to simulate the soil moisture component. It would be worth asking if the modelling could be made more efficient by constraining the soil moisture holding capacity routine rather with the SWI, which seems not to be particularly affected by the wet season effect, which was known in advance to be a limiting factor for the NDII.

**Answer:**

*We find that constraining the FLEX-SD models by the seasonal range and average NDII to distribute Sumax across 31 sub-catchments does not make the validation of $Su_i$ estimates by NDII an inappropriate approach. After all, we believe that the greater distinguishing powers between arid and productive sub-catchments resulting from the Su-NDII relationships is not invalid even though NDII was used in the estimation of $Su_i$.*

*Further, we do not think that SWI is a good constraint for distributed modelling, because it is partly model-based. It would lead to circular reasoning. NDII, however, is an independent observation. Therefore, using NDII as a constraint and using SWI as a validity check is a better approach.*

*We performed a formal split-series calibration-validation test and uncertainty analysis (requested by referee#1) of the different models which shows the added value of using NDII.*

*The fact that SWI works better during wet periods is not a real advantage. The soil moisture holding capacity is far stronger constrained by the stress periods, which is during the dry season. In the final check against the SWI we demonstrate that out approach also results in realistic values during wet periods and in wet ecosystems, such as evergreen forest.*

**Referee #2' s comment:**

Although in general the article seems well written to me, the wording of section 4.2 could be improved in order to avoid ambiguities (lines from 10 to 15 are not very straightforward). Tables 4 and 5 could also be improved, as they present duplicate data in some cases, and tabulations that do not seem appropriate. For example, by presenting results of the lumped FLEXL model calibrated for each of the various sub-catchments studied but tabulating these data under the table section: "for calibration at station P.1". Evidently a lumped model that was calibrated at P.1 could not provide results for individual upstream sub-catchments.

**Answer:**

*Section 4.2 has been revised accordingly. Tables 4 and 5 have been phased out and replaced with easily interpretable figures.*

**Referee #2' s comments:**

I have trouble endorsing some interpretations and statements expressed in the results section. But I think the article is especially lacking in its conclusions, which I believe are not being formally supported and proven. For example, on p. 14, L. 16-17, it reads: "The results indicate that the relationship with the average root zone soil moisture storage is affected by the ecology of the river basin.". The term "ecology" is extremely broad and complex, and there is no definition and characterization of it in the article. Even less, a scientific demonstration of such soil moisture-ecology relationship. The article goes on to state that: "The results confirm the power of NDII to capture the spatial variation of root zone soil moisture within the sub-catchment scale", which, again, is neither accurate nor was it strictly proven through the scientific method.

**Answer:**

*Misleading terms and phrases have been removed. Further, we have reinforced the benefits of NDII by elaborating on the consequential relationship between NDII-forced Sumax estimates and the percentage of sub-catchment forest cover, which is in itself a realistic finding.*

**Referee #2' s comment:**

On p. 15, L. 9-11, it reads: "As a result, the NDII appears to be useful to constrain hydrological models during dry conditions and both SWI and NDII appear to be useful to test model performance and to assess moisture states of river basins.". As stated earlier in this review, if NDII were used as a soil moisture model forcing, would not the simulation be expected to show this correlation? Furthermore, how sensitive is the model to forcing if even having constrained the soil moisture routine with the NDII, the result reveals a poor correlation between them during the wet season?

**Answer:**

*As per the prior answer (#2), the appropriateness of using average NDII to distribute Sumax across the 31 sub-catchments and deriving the flow and thereby $Su_i$ time series which were validated against both NDII and SWI is not unreasonable. The consequential variation in Sumax has produced $Su_i$ estimates which are increasingly sensitive to catchment aridity/productivity. We shall supplement this claim with Figure 9 in the manuscript, which reveals the influence of land-use composition on the $Su_i$, NDII and SWI signatures.*

**Referee #2' s comment:**
On p. 15, L. 18-19, it reads: "… it has been shown that it is required to account for the spatial variation of the moisture holding capacity of the root zone.". In fact, this was not shown, considering that the best independent validation tool presented in the article is the SWI, which was shown to be highly correlated with the simulated soil moisture storage regardless of the inclusion of the NDII constrains in the model. In addition, including or not the spatial variation of the moisture holding capacity of the root zone is not necessarily required in all cases, but rather depends on the objectives of the modelling exercise. For example, if the objective is the best possible calibration of a rainfall-runoff model in the outlet of a basin for seasonal hydrological prediction purposes, as we have seen, in some cases a lumped model could generate greater efficiencies.

**Answer:**
*As detailed in the discussion section, we attempted to convey that the quest for achieving strong Su-SWI across all sub-catchments is not the holy grail. Although this appears to be rather counterintuitive, it should be expected that sub-catchments with distinct $Su_i$ signatures (e.g., the forest-rich vs arid sub-catchments) correspond to SWI signatures to varying degrees. Therefore, the fact that the FLEX-SD-NDII$_{Avg}$ has appeared to exacerbate the variation of Su-SWI across the 31 sub-catchments should be perceived as the ability to better reflect heterogeneity.*

*Further, it is worth to be mindful of the objective at hand, which is to explore the potential of predicting flows in ungauged catchments, rather than comparing the performance of lumped and semi-distributed models at a gauging station. Inevitably, a calibrated lumped model would be expected to perform well at the outlet of the UPRB, if not better than the semi-distributed models. However, reframing the focus towards the great advantage that semi-distributed models have for ungauged basins (where flow and flood prediction is one of the main concerns of the hydrological community) and that the study found reasonable model performances (e.g., Figures 4 & 5) should be perceived as a valuable contribution.*

**Referee #2' s comment:**
P. 15, L. 20-21: "We concluded that the maximum of a series of annual ranges (NDIIMaxMin) and annual average (NDIIAvg) of NDII values offers an effective proxy for estimating the appropriate Sumax values in the different sub-catchments. ". Again, it seems to me that the data presented cannot support this conclusion.

**Answer:**
*The conclusion has been revised to accurately reflect the findings of the study. That is, the constraining by average NDII was appropriate.*

**Referee #2' s comment:**

P. 16, L. 1-2: "… However, during the wet season when soil moisture is replenished as a result of rainfall, NDII values are no longer well correlated with soil moisture.". In multiple parts of this article reference is made to "soil moisture" in order to later infer and conclude facts (such as its possible relationship with the NDII during the wet season), without the precaution that this soil moisture is not actually observed, but only a modeled value that was partially validated based on an indirect index (SWI). Due to the above, it does not seem appropriate to refer to soil moisture without specifying each time that it is a simulated value, nor drawing conclusions based on said simulated values, considering that they have not been sufficiently validated in the field and, therefore, cannot offer acceptable levels of accuracy and precision.

**Answer:**
*We have reiterated in fact that root zone soil moisture estimates are indeed simulated.*

**Referee #2' s comment:**

In conclusion, I believe that this article could be substantially improved by better contextualizing it within the current global research environment on the specific topic of using remote sensing to improve soil moisture simulation in distributed hydrological models (how many other research initiatives around the word are trying to use the NDII to better simulate soil moisture storage capacity?).

I would also suggest better organization, simplification and clarification of the description of the methodology (particularly section 4.2), and a thorough review of the modelling strategy, to avoid later interpretations that are irrelevant or based on spurious relationships (such as correlating an explicitly introduced forcing of the model outputs with those model outputs) or reaching conclusions that were not subject to hypothesis and testing (such as concluding about the ecology of a basin without having first systematized and analyzed that concept, or establishing the need to adopt a specific approach such as distributed modelling without sufficient empirical evidence to support it).

*Answer:*
*The manuscript now makes a remark to the opportunities in improving modern-day hydrological modelling through implementing remote-sensing techniques. The methodology has been revised to additionally include sections on model uncertainties (4.3) and relationship between the average root zone soil moisture storage ($Su_i$) and the average NDII and SWI (4.4). Further, the corresponding results to Section 4.2 has been divided into clearer steps (i.e., Sections 5.1.1-5.1.3).*

---

## Referee Report (RR1)

**Referee comment on the paper:**

**Using NDII patterns to constrain semi-distributed rainfall-runoff models in tropical nested catchments**

By Nutchanart Sriwongsitanon, Wasana Jandang, James Williams, Thienchart Suwawong, Ekkarin Maekan, and Hubert H.G. Savenije

I have finished my review of the paper "Using NDII patterns to constrain semi-distributed rainfall-runoff models in tropical nested catchments", by Sriwongsitanon et al., submitted to HESS. This paper outlines a comparison study of five models on the same set of nested catchments in Thailand – a lumped model (FLEXL) applied to individual gauges, the semi-distributed version of the same model (FLEX-SD), FLEX-SD modified by using NDII remote sensing metrics to inform the distribution of soil stores (models FLEX-SD-NDII$_{MaxMin}$ and FLEX-SD-NDII$_{Avg}$), and the independent semi-distributed URBS model. An attempt is made to demonstrate (1) the improved accuracy/realism of using NDII to inform the spatial distribution of soil stores while only calibrating a reference storage quantity and (2) the superiority of FLEX variants over the URBS model. This paper is very well-written, with interesting and appropriately prepared tables and figures, but it seems to me that there are two critical misinterpretations of some of the results. I consider that it would be acceptable for publication in HESS, subject to technical corrections on the following two approaches:

1) At multiple points in the paper, the authors report that the model "gained realism" (e.g., lines 29, 395, 397, 400), however, it is not until section 5.2 (lines 406-407) that some kind of objective definition of the term "realism" is given, since this is to be tested in the models by comparing the outputs of the models to observations of NDII and the global scale SWI dataset for verification. Furthermore, it is not until the conclusions section (lines 480-481) that the realism of the model parameters is directly associated with characteristics such as their distribution according to catchment characteristics comprising catchment area, reach length, and the NDII.

I believe that the methodological section should include a clear introduction of the definition that will be given in this article to the term "realistic", since the methods and definitions mentioned above are relatively subjective, since they are not supported by a statistical test or similar that allows drawing meaningful conclusions associated with a level of confidence about how true the values of the parameters (or model outputs) are, in relation to what we would scientifically consider their "true value" (which is usually achieved through methods such as statistical inference).

To be clearer, in my opinion, the article does not contain scientific evidence that the model gained realism in terms of its true values (parameters, results), but rather in terms of what the authors define as "realism" in lines 406-407 and 480-481 (which is valid as long as you make a formal definition of how the term will be used).

2) I think that both section 5.4.1 (Su-NDII relationship, this being an induced/forced relationship) and item 1 of the discussion section (exploring the causality between the aforementioned induced/forced relationship and the degree of aridity) should be deleted. The reason why is given in the first four lines of the discussion section. It makes no sense for me to present as a relevant finding that the NDII time series correlates well with Su values, considering that Su values were overtly and systematically constrained by NDII during the modelling exercise, and a marginal gain in the efficiency of the models was rather a trivial, expected result. Therefore, I do not believe that this particular modelling exercise

implies any scientific confirmation that the NDII is a "reasonable index to indicate root zone soil moisture during the dry season", if the only argument is the already expected higher correlation, which were in fact induced by procedure, between Su and NDII.

Again, concluding about signatures in catchments with various soil moisture capacities also seems methodologically inappropriate to me. If Su values from FLEX-SD-NDIIAvg (or any other NDII-based model) produce relatively higher NSE for sub-catchments with more evergreen forest, it is simply because the model forces these Su values to behave according to their corresponding NDII values, which in turn are directly affected by vegetation densities. Consequently, I would suggest removing all NDII time series from Figure 9 and others, and comparing only the simulated root zone moisture storage (Su) with SWI, which is in fact a relatively more independent spatiotemporal variable.

Despite all of the above, leaving aside the attempt of establishing a supposedly independent correlation between Su and NDII, and later use it to explain natural processes such as aridity or forest cover (which are the very factors underlying the NDII estimates and the forced modelling of Su), it would be interesting to try to explain why the NDII constrains Su so unevenly in sub-catchments with different percentages of evergreen forest. I think this exercise should conclude only on model structures, calibration methods, or uncertainty/sensitivity analysis of model parameters versus Su, rather than trying to provide a causal explanation for natural processes that have not been measured directly or statistically.

---

## Author Response (AR2)

*Referee comment on the paper:*

Using NDII patterns to constrain semi-distributed rainfall-runoff models in tropical nested catchments
By Nutchanart Sriwongsitanon, Wasana Jandang, James Williams, Thienchart Suwawong, Ekkarin Maekan, and Hubert H.G. Savenije

*The referee #1' s comments:*

I have finished my review of the paper "Using NDII patterns to constrain semi-distributed rainfall-runoff models in tropical nested catchments", by Sriwongsitanon et al., submitted to HESS. This paper outlines a comparison study of five models on the same set of nested catchments in Thailand – a lumped model (FLEXL) applied to individual gauges, the semi-distributed version of the same model (FLEX-SD), FLEX-SD modified by using NDII remote sensing metrics to inform the distribution of soil stores (models FLEX-SD-NDII$_{MaxMin}$ and FLEX-SD-NDII$_{Avg}$), and the independent semi-distributed URBS model. An attempt is made to demonstrate (1) the improved accuracy/realism of using NDII to inform the spatial distribution of soil stores while only calibrating a reference storage quantity and (2) the superiority of FLEX variants over the URBS model. This paper is very well-written, with interesting and appropriately prepared tables and figures, but it seems to me that there are two critical misinterpretations of some of the results. I consider that it would be acceptable for publication in HESS, subject to technical corrections on the following two approaches:

*Answer:*

*Thank you very much for your kindly comments and suggestions on the revised manuscript. We are truly appreciated your request to revise our manuscript to be suitable for publication in HESS as described in the followings.*

*The referee #1' s comments:*

1) At multiple points in the paper, the authors report that the model "gained realism" (e.g., lines 29, 395, 397, 400), however, it is not until section 5.2 (lines 406-407) that some kind of objective definition of the term "realism" is given, since this is to be tested in the models by comparing the outputs of the models to observations of NDII and the global scale SWI dataset for verification.

Furthermore, it is not until the conclusions section (lines 480-481) that the realism of the model parameters is directly associated with characteristics such as their distribution according to catchment characteristics comprising catchment area, reach length, and the NDII.

*Answer:*
*Thank you for the suggestion on using the term "realism". This has been referred to in the revised introduction as an objective to aim towards in this study. This helps to better back up the sentences where the term was mentioned in the original manuscript (i.e., in lines 395, 397, 400 of section 5.1.3 and lines 406-407 of section 5.2)*

*The referee #1' s comments:*

I believe that the methodological section should include a clear introduction of the definition that will be given in this article to the term "realistic", since the methods and definitions mentioned above are relatively subjective, since they are not supported by a statistical test or similar that allows drawing meaningful conclusions associated with a level of confidence about how true the values of the parameters (or model outputs) are, in relation to what we would scientifically consider their "true value" (which is usually achieved through methods such as statistical inference).

To be clearer, in my opinion, the article does not contain scientific evidence that the model gained realism in terms of its true values (parameters, results), but rather in terms of what the authors define as "realism" in lines 406-407 and 480-481 (which is valid as long as you make a formal definition of how the term will be used).

**Answer:**

*As aforementioned, a formal definition of model realism has been made in the introduction. This is reiterated in the final sentence of section 4.4 of the methodology. That is, the $R^2$ and NSE yielded for the Su-SWI and Su-NDII relationships serve as quantitative metrics for inferring model realism. The NSE values for both have also been added to Figure 9. We shall note that these values have always been intended as the key takeaway from our study – and we truly appreciate the feedback which has led to better clarification in our write-up.*

**The referee #1's comments:**

2) I think that both section 5.4.1 (Su-NDII relationship, this being an induced/forced relationship) and item 1 of the discussion section (exploring the causality between the aforementioned induced/forced relationship and the degree of aridity) should be deleted. The reason why is given in the first four lines of the discussion section. It makes no sense for me to present as a relevant finding that the NDII time series correlates well with Su values, considering that Su values were overtly and systematically constrained by NDII during the modelling exercise, and a marginal gain in the efficiency of the models was rather a trivial, expected result. Therefore, I do not believe that this particular modelling exercise implies any scientific confirmation that the NDII is a "reasonable index to indicate root zone soil moisture during the dry season", if the only argument is the already expected higher correlation, which were in fact induced by procedure, between Su and NDII.

Again, concluding about signatures in catchments with various soil moisture capacities also seems methodologically inappropriate to me. If Su values from FLEX-SD-NDIIAvg (or any other NDII-based model) produce relatively higher NSE for sub-catchments with more evergreen forest, it is simply because the model forces these Su values to behave according to their corresponding NDII values, which in turn are directly affected by vegetation densities. Consequently, I would suggest removing all NDII time series from Figure 9 and others, and comparing only the simulated root zone moisture storage (Su) with SWI, which is in fact a relatively more independent spatiotemporal variable.

Despite all of the above, leaving aside the attempt of establishing a supposedly independent correlation between Su and NDII, and later use it to explain natural processes such as aridity or forest cover (which are the very factors underlying the NDII estimates and the forced modelling of Su), it would be interesting to try to explain why the NDII constrains Su so unevenly in sub-catchments with different percentages of evergreen forest. I think this exercise should conclude only on model structures, calibration methods, or

uncertainty/ sensitivity analysis of model parameters versus Su, rather than trying to provide a causal explanation for natural processes that have not been measured directly or statistically.

**Answer:**

*As understood, in the development of our NDII-based FLEX-SD models, the NDII was used to constrain the root zone storage (Sumax) of each sub-catchment. However, we would like to emphasise that for FLEX-SD-NDII$_{Avg}$, only the average NDII value in each year was used to distribute the moisture capacities across the 31 sub-catchments. Similarly, for FLEX-SD-NDII$_{Max-Min}$, the maximum range of NDII over the 15-year time series was used.*

*After all, on the basis of previous findings from Sriwongsitanon et al. (2016), which said the "NDII is reasonable for indicating RZSM during the dry season", it is deemed a reasonable validation exercise to test the realism of our FLEX-SD-based models against this independent RS-based index (which detects canopy water content – Hardisky et al., 1983). For this reason, we do not see how constraining the FLEX-SD models by NDII and assessing the correlation between Su and NDII is unjustified, nor do we see any triviality in the improved model efficiency.*

*We perceive comparing and concluding about signatures across contrasting sub-catchments (i.e., Figure 9) to be methodologically meritable. The modelling exercise should not be perceived as favourable towards sub-catchments with more evergreen forests. As a matter of fact, the modelled Su signatures by FLEX-SD-NDII$_{Avg}$ yielded systematic increases in Su-NDII across the 31 sub-catchments (regardless of their vegetation densities). This notably improved realism over FLEX-SD owes to the distribution of Sumax values across these non-unique sub-catchments.*

*That being said, it is perhaps plausible that this modelling exercise does not explicitly confirm finding by Sriwongsitanon et al. (2016), that "NDII is reasonable for indicating RZSM during the dry season". However, it is perhaps better said that the constraining of semi-distributed models by NDII yields Su signatures which better reflects local characteristics.*

*Nonetheless, the discussion section of the manuscript has been carefully reviewed, and we are truly grateful for your feedback.*